# Vortex identification methods applied to wind turbine tip vortices

Rodrigo Soto-Valle[1], Stefano Cioni[2], Sirko Bartholomay[1], Marinos Manolesos[3], Christian Navid Nayeri[1], Alessandro Bianchini[2], and Christian Oliver Paschereit[1]

[1]Technische Universität Berlin, Hermann-Föttinger Institut, Müller-Breslau-Straße 8,10623 Berlin, Germany
[2]Università degli Studi di Firenze, DIEF, via di Santa Marta 3, 50139 Firenze, Italy
[3]City, University of London, SMCSE, Northampton Square, London EC1V 0HB, United Kingdom

**Correspondence:** Rodrigo Soto-Valle (rodrigo.soto@campus.tu-berlin.de)

**Abstract.** This study describes the impact of postprocessing methods on the calculated parameters of tip vortices of a wind turbine model when tested using Particle Image Velocimetry (PIV). Several vortex identification methods and differentiation schemes are compared. The chosen methods are based on two components of the velocity field and its derivatives. They are applied to each instantaneous velocity field from the dataset and also to the calculated average velocity field. The methodologies are compared through the vortex center location, vortex core radius and jittering zone.

Results show that the tip vortex center locations and radius have good comparability and can vary only a few grid spacings between methods. Conversely, the convection velocity and the jittering surface, defined as the area where the instantaneous vortex centers are located, vary between identification methods.

Overall, the examined parameters depend significantly on the postprocessing method and selected vortex identification criteria. Therefore, this study proves that the selection of the most suitable postprocessing methods of PIV data is pivotal to ensure robust results.

## 1 Introduction

The wake of a wind turbine is characterized by a massive presence of vortex structures. Two main types of concentrated vortices can be identified, which are shed from the root and the tip region, respectively. The latter form strong helical structures that influence the wake of the wind turbine.

The tip vortices are generated by the pressure difference between the top and lower side of the blade tip, which lead to a flow from the pressure side to the suction side of the blade (Karakus et al., 2008; Sherry et al., 2013b). In this way, the tip vortices of a wind turbine represent a source of energy loss (Shen et al., 2005) and noise (Arakawa et al., 2005). Moreover, the wake development needs proper consideration in the layout of a wind park (Marten et al., 2020), as it can affect the performance of wind turbines located downstream. Therefore, a more detailed characterization of the wind turbine wake vortices does represent a relevant research topic.

Since the first introduction of Particle Image Velocimetry (PIV) applied to wind turbine aerodynamics by Smith et al. (1990), many experimental investigations have been performed and a variety of methods have been employed to identify the vortex center and other characteristics. Yang et al. (2012b) studied the formation and evolution of helical tip vortices of a wind turbine

model under atmospheric boundary layer wind. A high variation of the position of the tip vortices is shown by using the vorticity in the identification. This effect is known as wandering or jittering and it is related to turbulence, vibrations of the model turbine (e.g., blades and tower) and the PIV system. Additional investigations (Maalouf et al., 2009; Soto-Valle et al., 2020) show the same effect using different identification methods such as the $Q$-criterion or circulation-based methods.

   Micallef et al. (2014) studied the mechanism of the formation of the tip vorticity on a wind turbine. The findings showed how
the vorticity is convected and forms a unique and symmetrical tip vortex behind the trailing edge. The location of the vortex center, identified by the maximum vorticity value, was found to be slightly inboard the rotor. In agreement with the previous finding, Xiao et al. (2011), by means of the vorticity, reported that the motion of the tip vortices moves first inward and then outboard of the rotor swept area, highlighting its importance in the aerodynamic modelling of the wake.

   Studies have also been carried out in water channels facilities. Sherry et al. (2013a) studied tip vortices from a submerged
wind turbine model. Results highlighted the breakdown of the wake due to the mutual interaction between helical structures of the tip vortices, which is highly dependent on the tip speed ratio. Additionally, the jittering of the tip vortices was also detected. Meyer et al. (2013) tracked the tip vortices from a wind turbine model using the vorticity magnitude. The procedure was done by choosing a reference vorticity magnitude, after visual inspection. Then, the location is estimated by averaging the positions where the vorticity magnitude is larger than the considered reference. Moreover, several studies rely on the identification of the
wind turbine tip vortices to assess retrofits such as winglets (Ostovan et al., 2018), rime ice effects (Jin et al., 2014) or surge motion impact (Fontanella et al., 2021).

**Table 1.** Wind turbine tip vortices studies employing the PIV technique and VIM details.

| contributor | test facility[a] | diameter [m] | N | VIM | scheme |
|---|---|---|---|---|---|
| Whale et al. (2000) | WC | 0.18 | 6 | vorticity magnitude | fifth order polynomial |
| Maalouf et al. (2009) | WT, closed-loop | 0.50 | 95 | circulation | integration |
| Xiao et al. (2011) | WT, open jet | 1.25 | n.s. | vorticity magnitude | n.s. |
| Yang et al. (2012a) | WT, closed-loop | 0.25 | 1000 | vorticity magnitude | n.s. |
| Micallef et al. (2014) | WT, open-jet | 2.00 | 100 | vorticity magnitude | central difference |
| Meyer et al. (2013) | WC | 0.38 | 100 | vorticity magnitude | n.s. |
| Sherry et al. (2013a) | WC | 0.23 | 25 | Graftieaux's method | solid-body rotation |
| Sherry et al. (2013b) | WC | 0.23 | 300 | swirling strength criterion | Richarson extrapolation |
| Ning and Yang (2013) | WT, open-jet | 0.25 | 960 | vorticity magnitude | n.s. |
| Jin et al. (2014) | WT, closed-loop | 0.15 | 300 | vorticity | n.s. |
| Ostovan et al. (2018) | WT, open jet | 0.94 | 1000 | zero induced velocity | central difference |
| Soto-Valle et al. (2020) | WT, closed-loop | 3.00 | 1200 | $Q$-criterion | central difference |
| Fontanella et al. (2021) | WT, closed-loop | 2.38 | 100 | vorticity magnitude | central difference |

(a) WT: wind tunnel, WC: water channel. n.s: not specified.

It is worth remarking that, once comparing the methods the inherent error introduced by the PIV technique must be accounted for. Table 1 includes the number of samples (or pair-samples in Stereo-PIV) used to analyze each contribution. The latter is a well-known parameter, directly related to the uncertainty level. This has been extensively used in literature to give a quantification of the uncertainty in PIV experiments (Grant and Owens, 1990; Micallef, 2012; del Campo et al., 2014; Micallef et al., 2016), Eq. 1 shows an example of the error in a measured velocity $u$ by Sherry et al. (2013b).

$$\epsilon_u = \frac{z\,I_u}{\sqrt{N}}, \tag{1}$$

Where $z$ is the confidence coefficient or critical value (normal distribution), $I$ is the turbulence intensity and $N$ is the number of samples. Moreover, it is overall agreed that actions to reduce uncertainty levels could be (1) the maximization of the number of samples to ensure repeatability and convergence of the results (Uzol and Camci, 2001; Ostovan et al., 2018); and (2) the use of subpixel algorithms (Scarano, 2001) giving errors below $0.1\,px$ (del Campo et al., 2014; Sciacchitano et al., 2013; Beresh, 2012; Fouras and Soria, 1998; Scarano, 2001).

Several vortex identification methods (VIMs) have been employed so far. However, consensus on the most suitable methodology for the study of vortices in the wake of a wind turbine has not been found yet, as shown in Table 1. Furthermore, upon examination of the literature, it is apparent that many studies do not provide the complete implementation methodology, such as the differentiation scheme, thus hindering an extensive comparison between methods.

This paper aims at comparing different vortex identification methods to evaluate their suitability to study specifically the tip vortices of a wind turbine. The methods are applied to velocity field planes that were obtained through PIV in the near wake of a wind turbine model located in a wind tunnel facility. Compared to previous investigations, the present study offers in a depth comparison, commonly used VIMs on the same wind turbine tip vortex measurement data set. The main goal is to identify similarities and differences of the methodologies, i.e., providing a direct insight into their application. Furthermore, a rigorous comparison of VIM application is provided, with the simultaneous study of six tip vortex parameters, namely: (1) streamwise location; (2) lateral location; (3) streamwise velocity; (4) lateral velocity; (5) core radius; and (6) jittering.

Thanks to the large number of analyzed samples, a statistical analysis is also included in order to give more insights into the challenges of each methodology. Three different VIMs are compared: vorticity, Q-criterion and Graftieaux. The first two VIMs require differentiation, thus, the application of six different schemes is examined. Moreover, Graftieaux's methodology is also tested in different scenarios. In this way, a total of 14 cases are presented, where each of the six parameters is investigated. This represents an important source of information to support future wind turbine tip vortices analyses in both experiments and simulations as the implementation is scalable and only requires velocity fields input

The following section, Sect. 2, gives the mathematical overview of the methods to identify vortices and the differentiation schemes applied on their implementation. Subsequently, the wind tunnel and test rig used to generate the experimental dataset are introduced in Sect. 3, followed by the methodology in Sect. 4. The results are presented in Sect. 5 to conclude with the most important remarks in Sect. 6.

## 2 Vortex identification methods

Many vortex identification methods have been proposed in the literature (Spalart, 1988; Hunt et al., 1988; Graftieaux et al., 2001; Vétel et al., 2010; Liu et al., 2016; Shkarayev and Kurnosov, 2017; Zhang et al., 2018a, b; Liu et al., 2020). In this work, three identification methods are compared. The chosen methods are based on the velocity field $U$, differing in their derivative orders. Consequently, the methods are based on the velocity field ($U$) and first-order derivatives ($\nabla U$). The selected methods are:

– Graftieaux's method, Graftieaux et al. (2001)

– Vorticity magnitude, Spalart (1988)

– $Q$-criterion, Hunt et al. (1988)

Additional methods can be derived from the eigenvalue analysis, such as $\lambda_2$, $\Delta$ or swirling strength criteria (Zhang et al., 2018b). However, for the scope of this research, they represent similar approaches and therefore, only the selected methods are analysed.

A full description of the selected methods is given below. For a more extensive review of VIMs, the interested reader is directed to Zhang et al. (2018b).

### 2.1 Graftieaux's method

This method identifies the vortex through a global quantity, $\Gamma_1$, from an equivalent solid-body rotation. This function allows to determine the location of the vortex center. Equation 2 shows the scalar, $\Gamma_1$, defined in a discrete space.

$$\Gamma_1 = \frac{1}{N} \sum_S \frac{(\overrightarrow{PM} \cdot \overrightarrow{U_M}) \cdot \overrightarrow{z}}{||\overrightarrow{PM}|| \, ||\overrightarrow{U_M}||} = \frac{1}{N} \sum_S sin(\theta_M), \tag{2}$$

where $P$ is a fixed point to evaluate, $\overrightarrow{U_M}$ is the velocity of the $M$ surrounding points to $P$ in the surface $S$, $\overrightarrow{PM}$ is the radius vector that connects the point $P$ with $M$. $N$ is the total number of points considered in the surrounding of $P$, and $\overrightarrow{z}$ is the unit vector, normal to the surface plane $S$. The angle $\theta_M$ is formed by vectors $\overrightarrow{PM}$ and $\overrightarrow{U_M}$.

Figure 1 shows a graphic representation of the parameters for the calculation of $\Gamma_1$. Over a two-dimensional frame, $\Gamma_1$ represents the topology of the surrounding flow to the point $P$. In this way, $\Gamma_1$ is the average contribution of the angles between the velocity $\overrightarrow{U_M}$ and the radius vector. Therefore, at the vortex center, the value of $\Gamma_1$ tends to be close to the unity because the velocity contribution is perpendicular to the radius vector.

Depending on the grid size, the value of $\Gamma_1$ might not reach unity as the center of the vortex could not be located on a grid point. Therefore, the vortex center is estimated as the position of the maximum value of $\Gamma_1$.

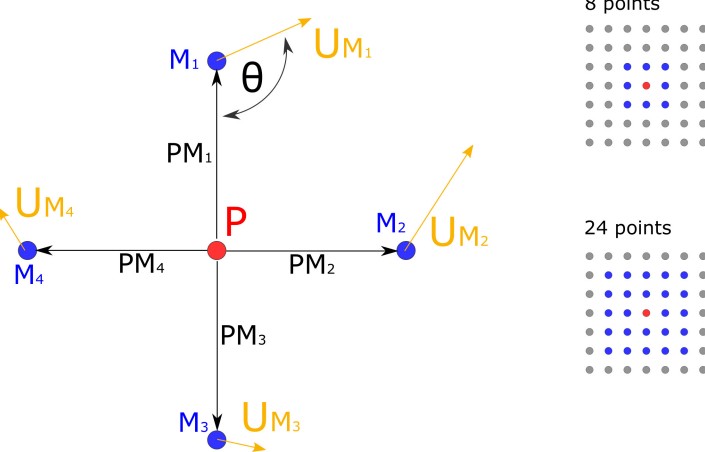

**Figure 1.** Sketch of Graftieaux parameters in $\Gamma_1$ implementation.

## 2.2 Vorticity magnitude

The vorticity is defined as the curl of the velocity field as show in Eq. 3

$$\omega = \nabla \times U \overset{2D}{=} \frac{\partial v}{\partial x} - \frac{\partial u}{\partial y}, \tag{3}$$

where $\omega$ is the vorticity. Equation 3 shows also a two-dimensional representation, where the out of plane vorticity is a function of the velocity field $U = U(u(x,y), v(x,y)) \equiv U(x,y)$.

In this way, the vorticity quantifies how the velocity vector changes when it moves in a direction perpendicular to it and therefore, is a natural candidate for vortex identification. Indeed, this method has been used for a long time (Spalart, 1988). In the vortex core, the vorticity is predominant compared to the shear rate deformation, due to the rotation of the fluid. Hence, the vortex core is identified as a region of high vorticity. At the vortex center, the vorticity reaches its maximum value. Consequently, the maximum value of vorticity can be used to locate the vortex center. However, as it has been pointed out by many authors (Liu et al., 2019; Zhang et al., 2018a), vorticity cannot distinguish between parallel shear motion and vortical motion. As an example, a laminar boundary layer shows high vorticity even though no vortical motion is present. Moreover, a vorticity threshold must be chosen to plot the vorticity iso-surfaces and the determination of the vortex core radius relies on other quantities such as the swirl velocity. Nevertheless, this method is based on first-order derivatives, which are easy to implement and commonly used in the literature.

## 2.3 $Q$-criterion

The most common vortex identification methods are based on the analysis of the velocity gradient $\nabla U$. For instance, from the analysis of the eigenvalues of $\nabla U$, Eq. 4, three invariants can be found ($P$, $Q$, $R$)

$$\lambda^3 + \lambda^2 P + \lambda Q + R = 0, \tag{4}$$

In particular, the second invariant $Q$ can be obtained through:

$$Q = \frac{1}{2}\left(tr\left(\nabla U\right)^2 - tr((\nabla U)^2)\right) \overset{2D}{=} -\frac{1}{2}\left(\left(\frac{\partial u}{\partial x}\right)^2 + \left(\frac{\partial v}{\partial y}\right)^2\right) - \frac{\partial u}{\partial y}\frac{\partial v}{\partial x}, \tag{5}$$

where $tr$ is the mathematical trace. In this way, the second invariant, Eq. 5, defines the $Q$-criterion. The method can be interpreted as the difference between the vorticity magnitude and the magnitude of the strain rate (Kolář, 2007). Hence, similar

to the vorticity magnitude, the vortex core will be characterized by positive large magnitudes of $Q$ since the rotation of the fluid is predominant compared to the strain rate in this region. In addition, areas characterized by parallel shear motion, without rotation, will not be identified as a vortex ($Q < 0$) and therefore, overcoming one of the limitations of the use of vorticity in vortex identification. Nevertheless, a threshold is still needed in order to determine the core region.

## 2.4 Differentiation schemes

As shown in the previous section, several vortex identification methods are based on the gradient of the velocity field, then inherently the evaluation of flow field derivatives is necessary. In this way, the differentiation of the velocity fields from either computational data or experimental techniques (as PIV) is needed. Both normally come in a discrete format.

In case of PIV data, the choice of the differentiation scheme becomes more relevant due to the presence of noise affecting the measurements. Noise sources include optical distortion, light sheet non-homogeneity, transfer function of CCD and particle

characteristics, among others (Foucaut and Stanislas, 2002). Indeed, the process of differentiation can amplify the effects and therefore compromise the results.

Many methods have been developed to calculate spatial derivatives from discrete data. The most frequently used methods are based on discrete differential operators applied to the surrounding points of the position to evaluate (Foucaut and Stanislas, 2002). In this way, the formulation of these schemes can be obtained through Taylor expansion. Equation 6 shows a general-

ization of the derivative scheme application on a function $f$ over the dimension $x$ in the point $j$ (Raffel et al., 2018).

$$\left.\frac{\partial f}{\partial x}\right|_i = f'_i + \sum_{p=n+1}^{\infty} \alpha_p \frac{\Delta x^{p-1}}{p!}\left.\frac{\partial^p f}{\partial x^p}\right|_i + \epsilon \frac{\sigma_f}{\Delta x}, \tag{6}$$

where $\Delta x$ is the grid spacing. The first term, on the right side, represents the implementation scheme (see Table 2). The following term is the truncation error, which depends on the number of elements from the Taylor expansion ($n$). Subsequently, the values of $\alpha_p$ are obtained through finite Taylor expansion. The last term on the right is the noise error (or uncertainty in

Table 2), $\epsilon$ is the noise amplification coefficient and $\sigma_f$ is the measurement noise level which could be estimated from the uncertainty from the measurements (Lourenco and Krothapalli, 1995). Accordingly to Foucaut and Stanislas (2002), there is a trade-off between the truncation error and the noise amplification, therefore, the increment of the order will increase the uncertainty of the scheme.

The backward, central and forward differencing schemes provide the simplest implementation. Nevertheless, additional

schemes have been studied with the purpose of either increasing the accuracy or reduce the uncertainty of the results. Raffel et al. (2018) presented, for example, the Richarson extrapolation which applied a fourth-order ($n = 4$) central differentiation scheme. Another tested methodology shown in the same work is the least-squares scheme, a second-order scheme, designed to minimize noise propagation. However, this approach has the tendency to smooth the estimation of the derivative because the outer data is weighted more than the inner data.

The latter schemes are defined for a single variable function and applied in one dimension at a time. Conversely, the velocity field can be influenced by the complete spatial coordinates. Therefore, the velocity gradient should depend on the surrounding flow. Raffel et al. (2018) proposed then, the circulation scheme that accounts for the effect of the surrounding flow (see Table 2). The first derivative is expressed as a central difference of derivatives in the other direction. This method reduces noise compared to the central difference scheme since the velocities of six neighboring points are considered instead of two.

Table 2 shows a summary of the mentioned schemes with their accuracy and uncertainty.

**Table 2.** Summary of differentiation schemes and implementation[a].

| Operator | Implementation $f_i'$ | Accuracy | Uncertainty, $\epsilon$ |
| --- | --- | --- | --- |
| Forward difference, FD | $\frac{f_{i+1}-f_i}{\Delta x}$ | $\mathcal{O}(\Delta x)$ | 1.41 |
| Backward difference, BD | $\frac{f_i-f_{i-1}}{\Delta x}$ | $\mathcal{O}(\Delta x)$ | 1.41 |
| Richarson extrapolation, RE | $\frac{f_{i-2}-8f_{i-1}+8f_{i+1}-f_{i+2}}{12\Delta x}$ | $\mathcal{O}(\Delta x^4)$ | 0.95 |
| Central difference, CD | $\frac{f_{i+1}-f_{i-1}}{2\Delta x}$ | $\mathcal{O}(\Delta x^2)$ | 0.7 |
| Circulation based method, CM | $\frac{f_{CD}|_{i,j-1}+2f_{CD}|_{i,j}+f_{CD}|_{i,j+1}}{4}$ | $\mathcal{O}(\Delta x^2)$ | 0.6 |
| Least squares, LS | $\frac{2f_{i+2}+f_{i+1}-f_{i-1}-2f_{i-2}}{10\Delta x}$ | $\mathcal{O}(\Delta x^2)$ | 0.32 |

(a) Composed from Raffel et al. (2018), Foucaut and Stanislas (2002) and van der Wall and Richard (2006).

## 3 Experimental dataset

The analysis shown in the rest of this paper relies on the stereo-PIV-dataset from previous work by Soto-Valle et al. (2020). In the following, the experimental setup is presented.

The experiments were carried out in the closed-loop wind tunnel at the Technische Universität Berlin. The wind turbine, Berlin Research Turbine (BeRT) (Pechlivanoglou et al., 2015), is a three-bladed, upwind horizontal axis wind turbine model. Blades are twisted, tapered, and based on Clark Y airfoil profile along the full span. Moreover, the blade-tip is sword-shaped and the Reynolds number, based on the circulation of the tip vortices, is $Re_\nu \approx 10^5$ (Soto-Valle et al., 2020). The freestream velocity and rotational frequency are fixed giving a tip speed ratio of $4.35$, which is the design-rated condition of the turbine.

The latter provides a constant operational condition to all the studied vortices. Table 3 reports details of the experimental setup.

The wind turbine model produces a $40\%$ blockage ratio in the wind tunnel, while this is quite relevant for performance measurements, it is thought to be acceptable for this study as all the identification methods and schemes are applied to the same dataset and with the focus of highlighting the differences in their outcomes. Therefore, conclusions should not be altered by this effect.

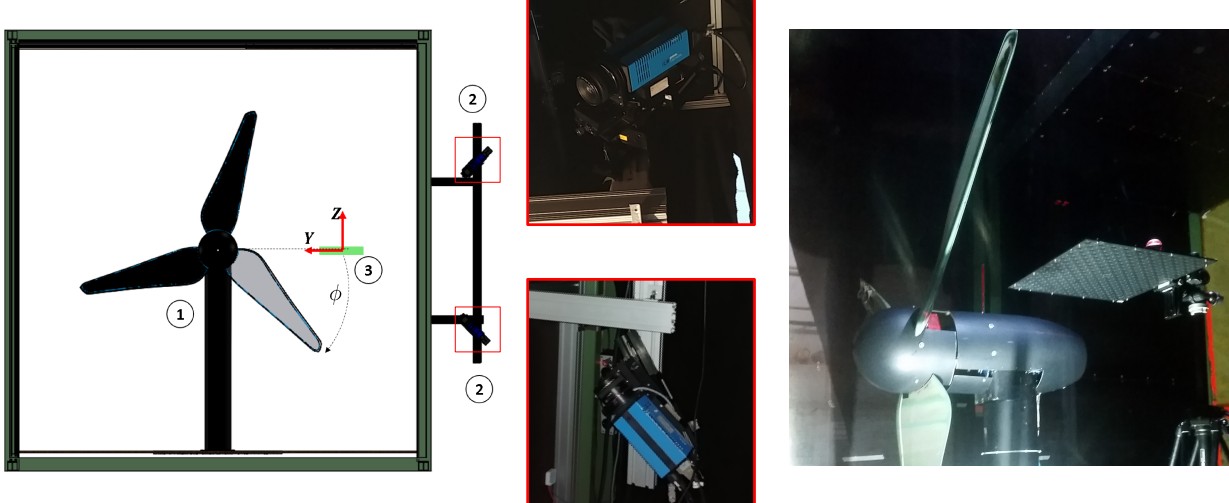

**Figure 2.** Front view sketch of Berlin Research Turbine ① (BeRT), cameras ② and laser sheet ③, left. Cameras system, middle. BeRT and calibration target in the test section.

The stereo-PIV system consisted of a Quantel Dual-Nd:Yag double laser with energy of $171mJ$, a mirror arm, the laser sheet optics and two cameras (CCD-chip). Additionally, an ILA synchronizer receives information of a reference blade azimuthal angle from a light sensor located in the nacelle. In this way, the phase-locked measurement is achieved by coupling the laser and blade position.

     The measurement plane was horizontal and was centered on the tip location when the blade was in the horizontal position. 180   In this study, only one vortex age is analyzed, $\phi = 40°$, consequently, all the studied parameters belong to the same vortex age, shed from consecutive rotations. A total of 1200 image pairs are recorded in the phase-locked position; this ensures enough information to obtain converged statistics of the results (Uzol and Camci, 2001; Ostovan et al., 2018). The image postprocessing is done with sub-pixel precision by the three-point Gaussian algorithm fit (Willert and Gharib, 1991) using the software PIVview3C (PIVTec GmbH). Table 3 provides details of the PIV system, while Fig. 2 shows a sketch of the facility 185   together with details of the camera and the calibration procedure.

**Table 3.** Operation and image acquisition details.

| operation parameters | | PIV parameters | |
|---|---|---|---|
| cross-section area | $4.2 \times 4.2 \ m^2$ | cameras | PCO 2000 |
| BeRT rotor radius | $1.5 \ m$ | lens focal length | $100 \ mm$ |
| blockage ratio | $40\%$ | resolution | $2048 \times 2048 \ px^2$ |
| freestream speed | $6.5 \ ms^{-1}$ | field of view | $435 \times 435 \ mm^2$ |
| rotational speed | $3 \ Hz$ | recordings | 1200 |
| tip speed ratio | 4.35 | laser pulse separation | $150 \ \mu s$ |
| turbulence intensity[a] | $3 - 6\%$ | interrogation window | $24 \times 24 \ px^2$ (50% overlapping) |
| phase-locked angle | $\phi = 40°$ | spatial resolution | $3.6 \ mm$ |

(a) reported by Bartholomay et al. (2017)

## 4 Methodology

A two-dimensional analysis is carried out on the dataset. To conduct a three-dimensional analysis of the vortex structures, additional parallel planes are needed. Therefore, only the two in-plane velocity components are used ($x - y$), even though the out of plane velocity $w$ is available from the stereo-PIV measurements.

The application to obtain the vortex properties follows, while the statistical analysis of the available data is described at the end of this section.

### 4.1 Vortex center

The velocity fields are analyzed through the application of the VIMs described in Sect. 2. In the case of the vorticity magnitude and Q-criterion, VIMs are implemented using the differentiation schemes shown in Table 1. In the case of Graftieaux's method, which does not use the derivatives of the velocity field, two amounts of surrounding points are considered (8- and 24-points, see Fig. 1).

Figure 3 shows a flowchart of the methodology used in this study. In this way, a dataset of velocity fields ($N = 1200$), from the PIV measurements is available. Both the average and the instantaneous flow fields are analyzed to identify the vortex center location, $(x_c, y_c)$, by means of the position of the maximum value of each VIM parameter ($\Gamma_1$, $\omega$, $Q$).

Results are presented in a normalized form and bounded by the unity. In the case of $\Gamma_1$, the parameter already fulfill these requirements by its definition, while the results of vorticity and $Q$ magnitudes are normalized according to Eq. 7:

$$\boldsymbol{\omega} = \frac{\omega}{\omega_{max}}, \boldsymbol{Q} = \frac{Q}{Q_{max}}, \tag{7}$$

where $\omega_{max}$ and $Q_{max}$ are the overall vorticity and $Q$ absolute maximum magnitudes. Moreover, the calculated distributions are presented in a reduced area of interest, as depicted in Fig. 4.

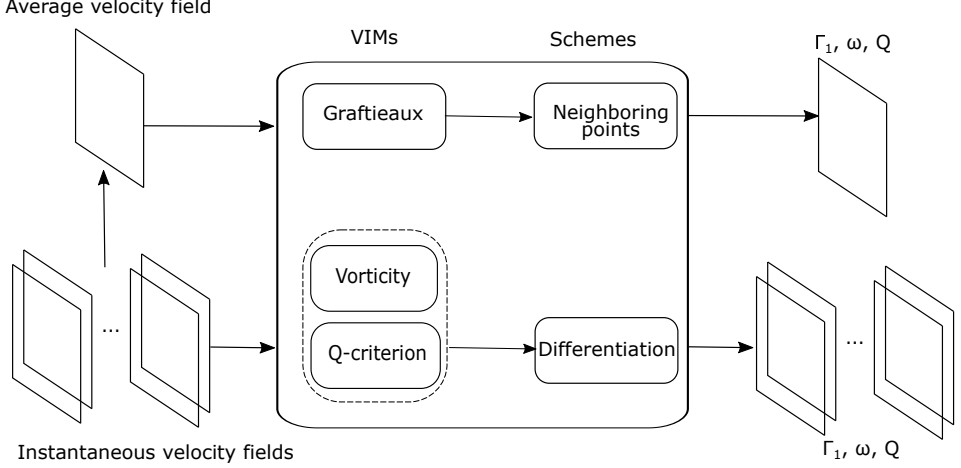

**Figure 3.** Flowchart of the implementation of the vortex identification methods and schemes.

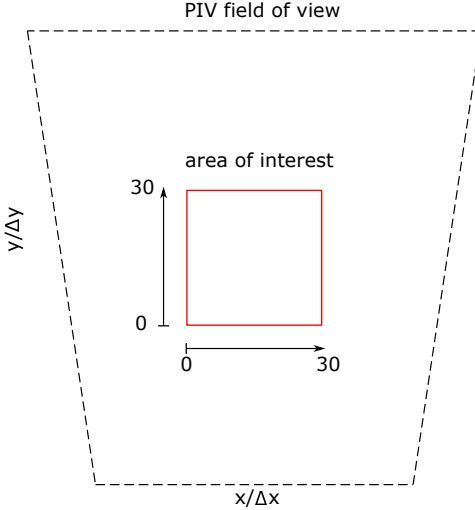

**Figure 4.** Field of view of PIV measurements and area of interest of the current study. Axes are normalized by the grid spacing.

## 4.2 Convection velocity

The tip vortex, after being shed, is both translating and rotating at the same time. Considering this, the convection velocity (downstream, $x$ and outboard directions, $y$) is estimated as the velocity magnitude corresponding to the vortex center location, Eq. 8. The latter is a common estimation in the literature (van der Wall and Richard, 2006; Yamauchi et al., 1999) and it has the advantage that only one vortex age is needed. However, the estimation is also affected by both the VIM and the scheme chosen on their application.

$$u_c = u(x_c, y_c); \; v_c = v(x_c, y_c).\tag{8}$$

## 4.3 Core radius

The core radius is calculated using the following steps:

1. The induced velocity field $U'$, Eq. 9, is obtained by means of the subtraction of the convection velocity from the velocity field. The resulting velocity field is characterized by presenting the induced contribution only.

$$U'(x, y) = U(x, y) - U(x_c, y_c).\tag{9}$$

2. The swirling velocity is analyzed through vertical and horizontal lines using the vortex center as an origin reference. The study is done in both directions to check the symmetry of the vortex, as vortices can have asymmetric shapes (Skinner et al., 2020).

$$U_{\theta,x} = U'(x, y = y_c); \quad U_{\theta,y} = U'(x = x_c, y).\tag{10}$$

3. A spline line is fitted to the swirling velocity curves. The radius, $r_c$, is estimated as half the distance between the maximum values of the fit curve.

The procedure is repeated for each VIM and scheme and applied to both the average and the instantaneous velocity fields. Figure 5 shows the instantaneous induced velocity, $v(x, y)$ of a representative PIV velocity field and the corresponding swirling velocity.

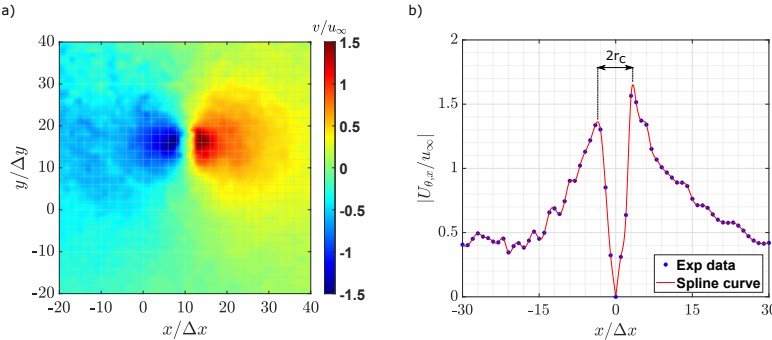

**Figure 5.** a) Induced velocity field $v'(x, y)$ . b) Swirling velocity with the x-axis shifted to the corresponding vortex center.

Both vortex characteristics, convection velocity and core radius, are normally used to describe the evolution of the vortex at different ages (Snel et al., 2007; Nilsson et al., 2015) as well as a parameter to quantify induced drag penalties from the tip vortices (Ostovan et al., 2018).

## 4.4 Statistical analysis

A statistical analysis is made over the complete dataset of the instantaneous velocity fields. In this way the vortex center, the convection velocity and the core radius are analyzed in terms of their location and magnitude variations.

An ellipse is used to define the jittering characteristic zone, similar to the work of Sherry et al. (2013b). The semi-axes of the ellipse $a$ and $b$ are defined to include all vortex center locations and the overall surface of the ellipse is calculated as $S = \pi ab$.

## 5 Results and discussion

The results are presented as follows. First, an overview of the VIMs and schemes applied to the average velocity field are shown using the vortex center locations and convection velocities based on the three identification parameters $\Gamma_1$, $\boldsymbol{\omega}$, $\boldsymbol{Q}$. The subscripts provide the information about the scheme implementation, e.g, $\boldsymbol{\omega}_{CD}$ shows the results from vorticity using the central difference scheme.

Second, a statistical analysis of the complete set of instantaneous velocity fields is performed. In this way, the location of the vortex center locations is studied in order to define the shape, distribution and surface of the jittering zone described by each VIM and scheme. Additionally, these results are used to compare the scattering of the convection velocities and core radii.

### 5.1 Average velocity field

Figure 6 shows the magnitude distribution of the parameter $\Gamma_1$ after the application of Graftieaux's method with 8- and 24- points on the average velocity field. Both cases show a concentration of the magnitude in a core with one peak in the same location at $(x/\Delta x, y/\Delta y) = (11, 14)$ and with an almost identical magnitude of $\Gamma_1 \approx 0.97$, although the $24-$p scheme extends its distribution over a larger zone.

Graftieaux's method has been developed for stationary vortices, while indeed, the test case is a superposition of the vortex-induced velocities and the streamwise flow, which convects the vortex downstream. The latter difficulty is overcome by subtracting the background velocity. Sherry et al. (2013b) proposed subtracting the average phase-locked velocity, $\overline{\overline{U}}$ obtained by averaging each magnitude, streamwise ($u$) and lateral ($v$) from the full field of view.

Figures 7 and 8 depict the results from the vorticity and Q-criterion, respectively. In contrast to $\Gamma_1$, The application of $\boldsymbol{\omega}$ and $\boldsymbol{Q}$ VIMs is not dependent on the subtraction of the background flow, so this step is not performed for these two methods.

Vorticity and Q-criterion provide almost identical results with a concentrated region defining the vortex, as apparent in Figs. 7 and 8. Nevertheless, in case of BD, CD, FD and RE schemes (Figs. 7 and 8, a-d) the vortex cores do not exhibit a unique peak in their center. Instead, the vortex is characterized by high radially distributed magnitude and multiple local maxima can be identified ($\boldsymbol{\omega} \geq 0.8$). Furthermore, in the center of the core, the magnitude is lower than the perimeter. The scheme cases LS and CM (Figs. 7 and 8 e-f) exhibit a smoother distribution compared to the other schemes. A unique maximum is found closer to the center of the vortex core in each case.

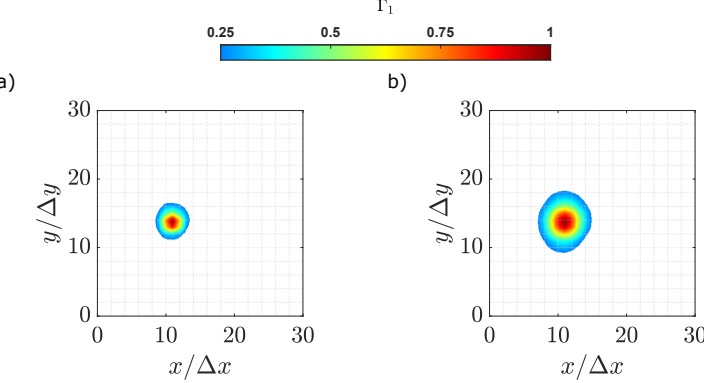

**Figure 6.** $\Gamma_1$ magnitude on the area of interest, where the methodology is applied after removing the average velocity from the field of view. a) $8-$points. b) $24-$points.

The presence of multiple maxima and the ring-like distribution of the $\omega$ and $Q$ parameters can be explained through different hypotheses. On the one hand, the cause could be the level of noise in the vortex core because of the lack of seeding (Foucaut and Stanislas, 2002; van der Wall and Richard, 2006). The rotational motion of the fluid causes the seeding particles to be pushed at the edges of the vortex. For this reason, the velocity vectors shall be evaluated through interpolation, introducing a further source of uncertainty in the results. In this way, the contours of $\omega$ and $Q$ have a single peak concentration for the schemes with the lowest uncertainties (LS and CM) while two peak concentrations appear for the schemes with higher uncertainty (CD, RE, BD and FD).

On the other hand, the presence of multiple maxima might also be due to small-scale structures within the vortex, as suggested by Bonnet (1996). It is conceivable that these structures might originate during the shedding of the tip vortex from the blade. Certainly, the pressure difference between the pressure and suction sides of the blade is only one of the many effects that take part in the formation of the tip vortices. Several experiments show that the flow at wingtips involves the interaction of multiple vortices, shear layer instabilities, flow separation and re-attachment (Giuni and Green, 2013; Devenport et al., 1996; Micallef, 2012). The involved structures are also affected by the blade shape, tip geometry, Reynolds number, and load distribution (Giuni and Green, 2013) and generally merge into a single structure. In conclusion, the multiple peaks could be caused by the uneven shedding of vorticity in the chordwise direction. In the work of Micallef et al. (2014), a study of the formation of the tip vortices in a horizontal axis wind turbine, a complex vorticity distribution along the blade chord is observed, which seems to cause multiple vorticity peaks inside the core. These multiple peaks can be identified in the vortex core even after the tip vortex has been shed from the blade. In present results, the same effect is obtained when the high uncertainty schemes are applied (CD, RE, BD and FD).

Among these hypotheses, the first one seems the most suitable. It is possible that artifacts are produced on some of the schemes applied, where the concentration of seeding is diminished. These artificial peaks are not present in the results using the Graftieaux method because the methodology includes information from a larger amount of grid points.

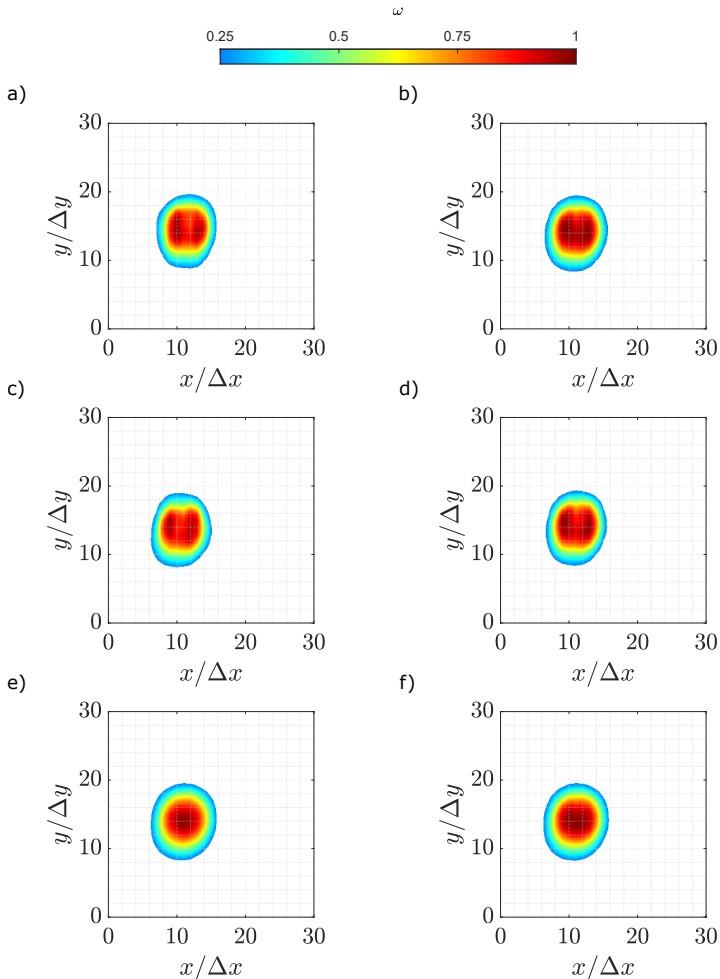

**Figure 7.** Normalized vorticity magnitude on the area of interest. a) Backward difference. b) Central difference. c) Forward difference. d) Richardson extrapolation. e) Least square. f) Circulation method.

In fact, eight and 24 points are employed to estimate the parameter $\Gamma_1$. In the case of BD, CD and FD only two grid points are considered. RE and LS schemes use four grid points, with the difference that in the first case the inner points are considerably weighted more (see Table 2); the opposite happens for the LS case. CM scheme employs six grid points. Therefore, either weighting more the outer part of the derivative estimation (LS scheme) or considering more grid points (CM scheme) contribute to repairing the artifacts and also evidence that the issue only occurs in the inner part of the vortex.

Regarding the position of the vortex centers, the locations are shown in Fig. 9, overlapped with the vorticity magnitude distribution. It can be seen that the identification method does not have a strong influence on the estimation of the vortex center location with differences up to $y/\Delta y = 2$ and $x/\Delta x = 4$ grid steps in the lateral and streamwise directions, respectively.

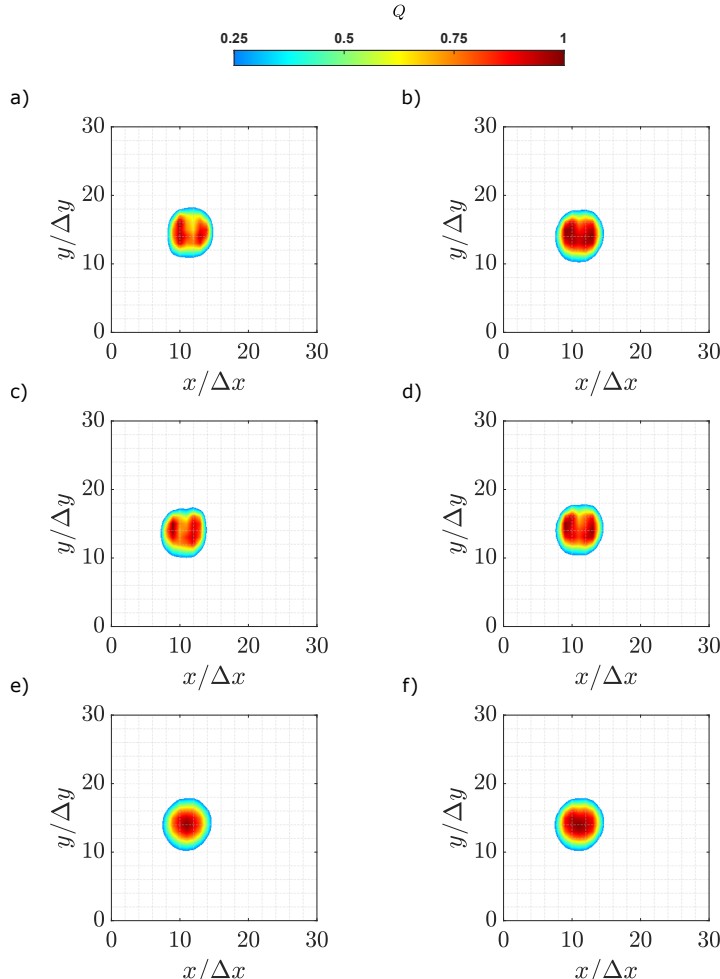

**Figure 8.** Normalized Q-criterion magnitude on the area of interest. a) Backward difference. b) Central difference. c) Forward difference. d) Richardson extrapolation. e) Least square. f) Circulation method.

In case of vorticity and Q-criterion employing BD, CD, FD and RE schemes, the estimation of the vortex center is different from the geometrical center of the shape described by the core. As a result the convection velocity also differs significantly between schemes. Figure 10 illustrates the axial and lateral velocities estimated from each VIM and scheme.

    Vorticity and Q-criterion provide the same velocity magnitudes, when the same scheme is applied. In case of CD, LS and CM schemes, the streamwise velocity magnitudes are in good agreement with previous results from Soto-Valle et al. (2020),
where the conditional average methodology (van der Wall and Richard, 2006) was implemented with a resulting ratio of $u_c/u_\infty = 0.85$. Conversely, the schemes BD, FD and RE present scattered velocities on both axial and lateral directions. In case of the Graftieaux's VIM, it is independent on the number of neighboring points and the magnitude is close to the smoothest schemes from the other VIMs.

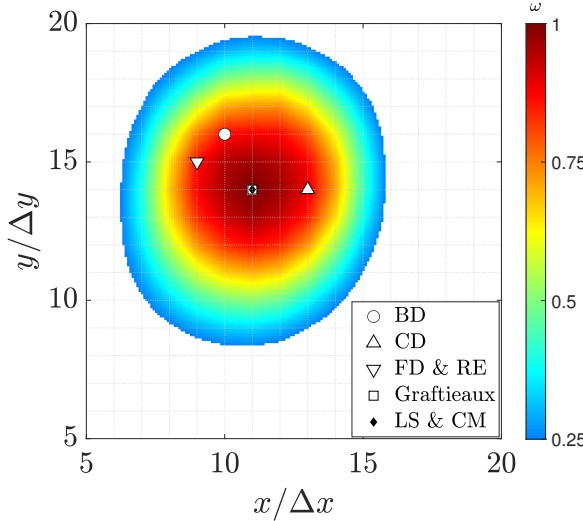

**Figure 9.** Vortex center locations for different differentiation schemes. The vorticity magnitude contour based on the least squares scheme is shown.

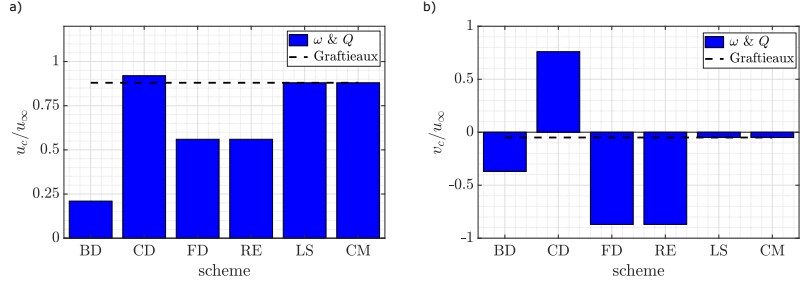

**Figure 10.** Average convection velocities. a) streamwise velocity. b) lateral velocity.

Therefore, the estimation of the convection velocity is recommended with the smoother VIMs and schemes i.e., the Graftieaux method or vorticity and Q-criterion while employing LS or CM schemes. Additionally, since there is a low scattering in vortex locations among VIMs and schemes, the convection velocity can be alternatively calculated by comparing several vortex locations over time, fitting streamwise and lateral locations separately (Snel et al., 2007; Soto-Valle et al., 2020). However, more than one vortex age is needed.

Based on selected operational parameters (see Table 3), the uncertainty of the velocity magnitudes is below $\pm 0.4\%$ of the freestream velocity (see Eq. 1), which is equivalent to $0.026\ ms^{-1}$ with a $98\%$ confidence. This uncertainty level does not affect the location of the vortex centers of the averaged velocity field or the other studied parameters, as they rely on the vortex center location. For completeness, a statistical analysis of the instantaneous velocity fields is done and is presented in the following section.

## 5.2 Statistical analysis

The vortex center locations are identified on each instantaneous velocity field, which constitutes the complete PIV dataset. For readability, only the Graftieaux 24-points and vorticity VIMs are presented. The reason is due to Graftieaux 8-points and Q-criterion present small differences with the aforementioned cases. For completeness, the full set of results can be found in App. A.

Figure 11 a) shows a contour diagram of the tip vortex center positions obtained through the Graftieaux VIM and the 24-
points scheme applied to each velocity field. In fact, the zone can be highlighted as an ellipse that has its main axis on the lateral direction with five grid points more than the streamwise direction. Figure 11 b) shows the added up distribution along the streamwise direction.

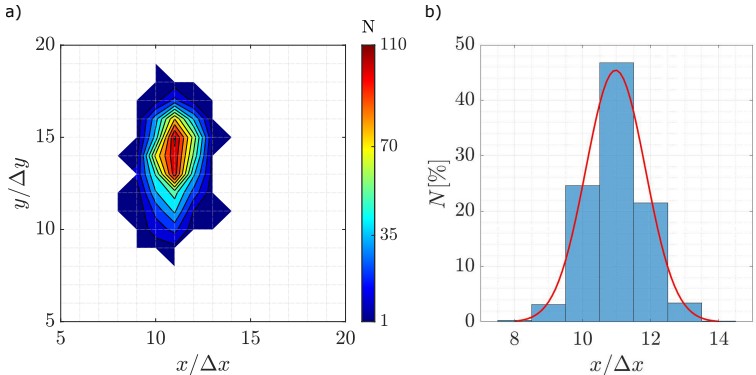

**Figure 11.** Jittering evaluation using Graftieaux VIM with 24-points scheme. a) contour distribution of the vortex center locations over the area of interest. b) Probability distribution among the streamwise direction.

From Fig. 11, the jittering effect is clearly visible and agrees with previous results from fixed wings (Thompson, 1983; Giuni and Green, 2013; Bandyopadhyay et al., 1991; Beresh et al., 2010), helicopter rotors (van der Wall and Richard, 2006;
Mula et al., 2011) and wind turbines (Maalouf et al., 2009; Soto-Valle et al., 2020; Sherry et al., 2013a). According to these references, the source of the jittering can be varied such as geometry effects, wall boundary layer turbulence, free stream turbulence, surface irregularities or changes in the core structure. Additionally, the vibrations of either the model or the test rig supports can produce small changes in the field of view, resulting in the meandering motion of the vortex. In this way, the jittering in Fig. 11 a) shows the spreading of the vortex center locations with $y$ as a prevalent direction. The probability
distribution over the streamwise direction, Fig. 11 b), fits very well with a normal distribution. These characteristics are in agreement with Sherry et al. (2013a), where it is shown that the jittering of the tip vortices in the wake of a horizontal axis wind turbine is predominant in the radial direction compared to the streamwise direction and that at early vortex ages, the normal distribution is a good fit of the tip vortex center distribution. In agreement with the results presented here, Mula et al. (2011) also observed that the jittering of tip vortices generated by helicopter rotors present a preferential direction.

Figure 12 shows the jittering zones for the Grafieaux method and vorticity calculations with the different differentiation schemes. For the purpose of clarity, only ellipse perimeters that contains the $100\%$ of the vortex center locations are presented. It is noticed that the ellipse described by the Grafieaux VIM is thinner in the streamwise direction observing approximately two to four grid steps less than the vorticity VIM, depending on the schemes applied. In fact, the area described by the ellipses on the Grafieaux VIM is $80\pm6$ while in the case of the vorticity and $Q$ are between $145\pm24$, corresponding to an $80\%$

increment. The size variation between schemes in the vorticity VIM, Fig. 12, is more uniform in terms of directions and goes approximately to one grid step in any direction.

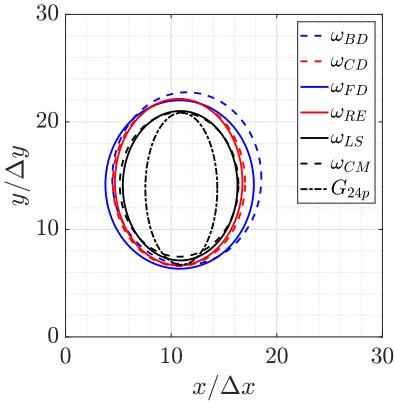

**Figure 12.** Jittering zones over the area of interest with Grafieaux and vorticity VIMs.

     Even though the area swept by the scattering of the estimated vortex center locations are similar in magnitude, in the case of vorticity and $Q$, the probability distribution over these zones differs between the applied schemes. Figure 13 shows the contour diagram of two representative distributions applying vorticity VIM. Hence, Fig. 13 b), which is done with LS scheme, shows

a more concentric distribution than CD scheme,Fig. 13 a). Q-criterion and the rest of the schemes exhibit similar results (see App. A).

     Figure 14 shows the added up distribution towards the streamwise and lateral axes when the vorticity VIM and the CD scheme are applied, together with fitting curves. Each fitting curve is chosen using higher the coefficients of correlation ($R^2$) between normal, binormal and Weibull distributions. In Fig. 14 a), it can be noticed that the spreading of the vortex centers is

over two peaks with a distance of approximately $4$ grid points and therefore, a binormal distribution fits better with the data with a coefficient of correlation of $R^2 = 0.99$. In case of the lateral direction, Fig. 14 b), a binormal and Weibull distributions exhibit good fitting, with the coefficients of correlations of $R^2 = 0.97$ and $R^2 = 0.95$, respectively.

     Figure 15 shows the best-fit curves of the probability distribution when the differentiation schemes BD, CD and LS are applied. In the streamwise direction, Fig. 15 a), the LS scheme presents a normal distribution as the best fit. This is in agreement

with the unique maximum observed in the analysis of the average velocity field. Instead, for all the other schemes the binormal distribution is the best fit. The difference is due to the smoothing properties of the LS scheme. In fact, the application is

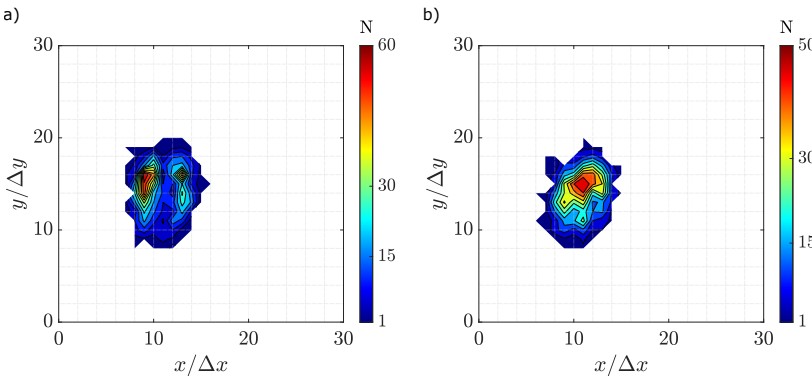

**Figure 13.** Contour distribution of the vortex center locations over the area of interest from vorticity VIM. a) Central difference scheme. b) Least square scheme.

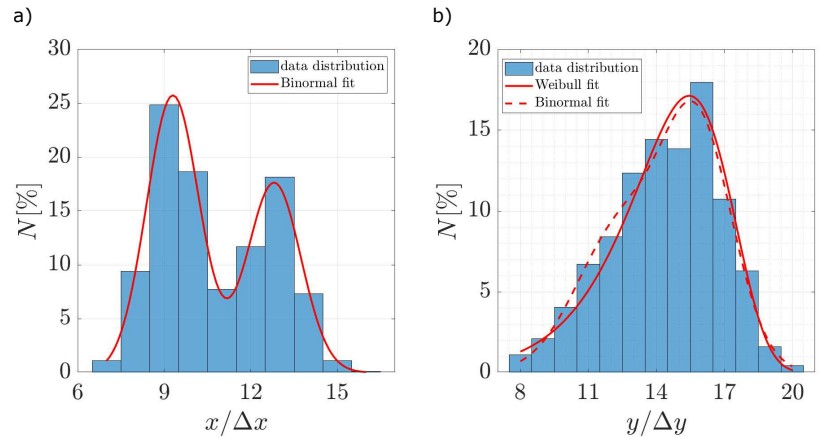

**Figure 14.** Probability distribution of the vortex center locations using vorticity VIM and CD scheme. Additionally, red lines show fitting curves. a) Streamwise direction. b) Lateral direction.

implemented using points up to the two positions farther in the grid, where the outer points are weighted more than the inner ones (see Table 2).

Similarly, in the lateral direction, Fig. 15 b), the binormal distribution is likely the best fit with a coefficient of correlation between $0.97 - 0.99$. In this case, peaks are very close, which also gives the Weibull distribution a good fit with coefficients of correlation between $0.95 - 0.97$. In this direction, only one position is concentrated, consequently, the smoothing effect produced by the LS scheme does not have the same effect in this direction.

Overall, the estimation of the vortex center location is influenced by the VIM and scheme. In the same way, the convection velocity and the core radius are affected by the implemented methodology.

To see the effect of the VIM and schemes on the instantaneous convection velocity, Figure 16 shows the normalized convection velocity in the streamwise direction, $u_c/u_\infty$. Each black dot represents the convection velocity estimated through the

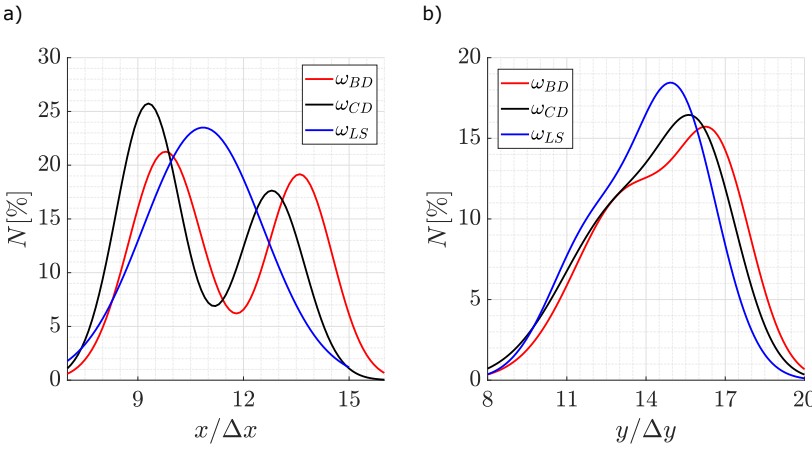

**Figure 15.** Probability distribution of the vortex center locations while the vorticity VIM is applied using BD, CD and LS schemes. a) Streamwise direction. b) Lateral direction.

corresponding VIM and scheme on each velocity field from the PIV dataset. At the same time, the average of the dataset magnitudes is visible as a solid blue line. Moreover, the average value obtained when the methodology was applied on the average velocity field (see Fig. 3 and Sect. 5) is displayed with a solid red line.

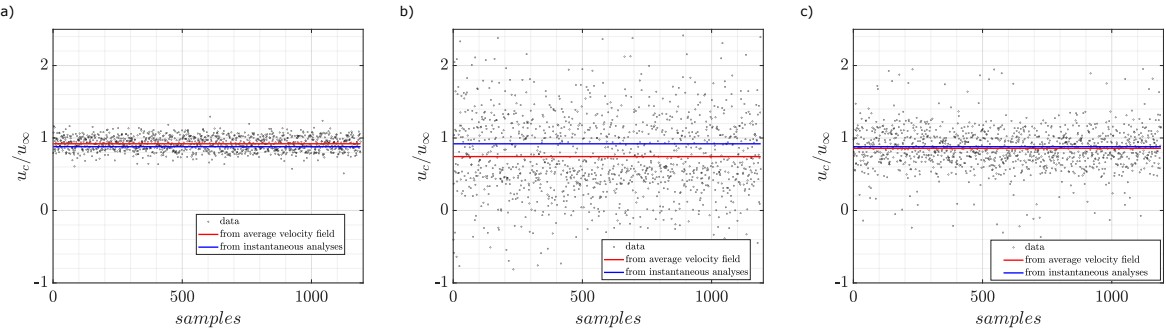

**Figure 16.** Normalized convection velocity in the streamwise direction. a) Graftieaux. b) Vorticity magnitude, central differentiation scheme. c) Vorticity magnitude, least square scheme.

Graftieaux VIM, Fig. 16 a), presents a variation of $10\%$ around its average value (blue line). In the case of the vorticity VIM, Figs. 16 b) and c), the variation increases up to $70\%$ and $33\%$ for the CD and LS schemes, respectively.

Several estimations fail, such as $u_c/u_\infty < 0$ or $u_c/u_\infty > 1$, as well in the average velocity analysis, due to the fact that they are located at the edges of the vortex, i.e., they are highly affected by the induced velocities of the vortex. Figure 17 shows one representative case of instantaneous vorticity where the central scheme is applied. It can be noticed that the highest magnitudes 370    of vorticity are spread from the center of the quiver lines.

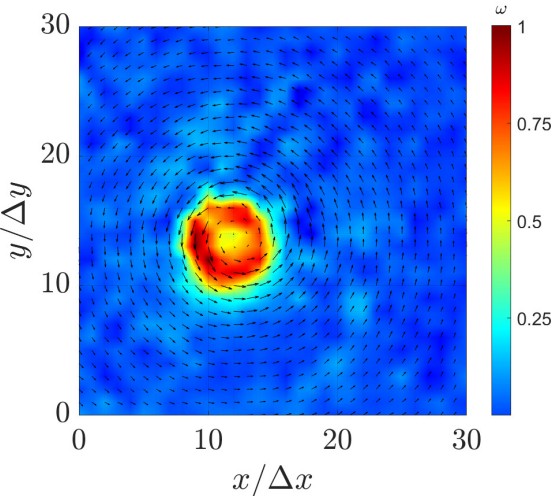

**Figure 17.** Instantaneous vorticity magnitude with the central differentiation scheme and quiver lines of the velocity field.

However, in the case of Graftieaux, vorticity and Q-criterion (schemes $CD$, $LS$ and $CM$), this effect is not prominent and average results have less than $10\%$ error compared with previous estimation $u_c/u_\infty = 0.85$ (Soto-Valle et al., 2020), obtained through the conditional averaging methodology. This error difference increases between $12 - 18\%$ in the case of vorticity and Q-criterion (schemes $CD$, $FD$ and $RE$) and to more than $50\%$ with the scheme $BD$.

The visible ring-like concentration in Fig. 17 contrasts the uneven shedding hypothesis previously formulated in the average results, supporting the idea of an artifact of the schemes, as it preserves the same ring-like structure even when an instantaneous velocity field is analyzed.

  Table 4, shows the full set of average values and their corresponding standard deviations ($SD$). As discussed, vorticity and Q-criterion yield similar results. Nevertheless, the $SD$ results from Q-criterion are equal or higher than vorticity in all the
schemes and in both directions, which is presumable caused by the power of two on its derivatives implementation.

  The average convection velocity is also estimated from the average flow field (red line). However, It should be noted that average results must not be overstated because of the tip vortex jittering (van der Wall and Richard, 2006). It is also remarkable how convection velocity has the lowest results from BD scheme and on the contrary, the highest from FD scheme. In fact, both schemes ignore information either forward or backward from the grid on the implementation of differentiation. Based on the
above and due to the large values of $SD$ after the application of these schemes, they are not recommended for vortex analysis.

In case of the vortex core radius, even when all the scattering of the convection velocity influences its calculation (see Sect. 4.3) the variation of the radius varies by just a couple of grid steps. In this way, the average values are similar between VIMs and

**Table 4.** Convection velocity magnitudes from the average velocity field and the statistics after instantaneous velocity field analyses.

| VIM | scheme | From average velocity field analysis | | From instantaneous velocity field analyses | | |
| --- | --- | --- | --- | --- | --- | --- |
| | | $u_c/u_\infty$ | $v_c/u_\infty$ | $\overline{u_c/u_\infty}$ | $\overline{v_c/u_\infty}$ | $SD$ |
| Graftieaux | 8-p | 0.88 | -0.05 | 0.92 | -0.02 | 0.10-0.13 |
| | 24-p | 0.88 | -0.05 | 0.92 | -0.02 | 0.10-0.13 |
| Vorticity | BD | 0.21 | -0.37 | 0.38 | 0.46 | 0.69-0.90 |
| | CD | 0.92 | 0.76 | 0.74 | -0.15 | 0.53-0.68 |
| | FD | 0.55 | -0.87 | 0.95 | -0.41 | 0.64-0.93 |
| | RE | 0.55 | -0.87 | 0.72 | -0.16 | 0.59-0.68 |
| | LS | 0.88 | -0.05 | 0.86 | -0.13 | 0.28-0.48 |
| | CM | 0.88 | -0.05 | 0.82 | -0.14 | 0.41-0.65 |
| Q-criterion | BD | 0.21 | -0.37 | 0.4 | 0.44 | 0.70-0.92 |
| | CD | 0.92 | 0.76 | 0.75 | -0.16 | 0.58-0.72 |
| | FD | 0.55 | -0.87 | 1.0 | -0.37 | 0.65-0.95 |
| | RE | 0.55 | -0.87 | 0.73 | -0.14 | 0.62-0.74 |
| | LS | 0.88 | -0.05 | 0.86 | -0.13 | 0.28-0.48 |
| | CM | 0.88 | -0.05 | 0.83 | -0.15 | 0.42-0.68 |

schemes. Figure 18 shows the normalized core radius, $r_c/\Delta x$ when it is estimated through along the horizontal axis. It is found that the results on the vertical axis are slightly higher, by $0.3\Delta x$, which is small enough to assume the vortex as symmetric.

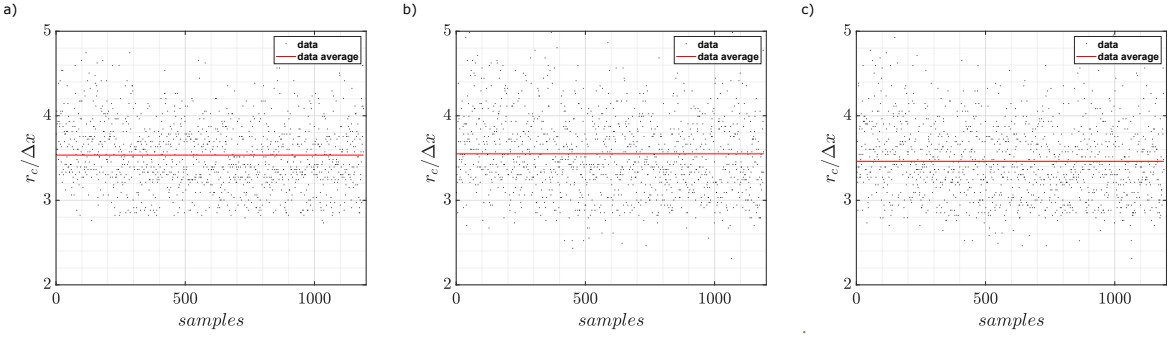

**Figure 18.** Normalized core radius. a) Graftieaux. b) Vorticity magnitude with CD scheme. c) Vorticity magnitude with LS scheme.


In this way, the magnitude of the core radius is $r_c/\Delta x \approx 3.4$ independently on the VIM and scheme applied. The standard deviation varies between methods; however, it is always lower than $1\Delta x$. This is expected due to the small number of grid points that are found within the core (see Fig. 5). Indeed, the results from the average velocity field are also similar between VIMs and schemes but slightly higher than the statistical results, $r_c/\Delta x \approx 4.4$.

## 395   6   Conclusions

Several Vortex Identification Methods (VIMs) and implementation schemes have been applied to the two components of the velocity field data in the near wake of a wind turbine model, obtained through PIV measurements. The methodology was applied to the average velocity field as well as the instantaneous velocities resulting in a statistical analysis of the PIV dataset.

In case of the average flow field, the chosen VIMs and schemes provide different magnitude distributions of the identification
parameters. Nevertheless, the vorticity and Q-criterion yield the same estimations of the vortex center locations in all the schemes analyzed. Hence, as long as the vortex is well-formed the vorticity VIM is preferred over the Q-criterion because of the lower standard deviation results.

Through the statistical analysis, it is concluded that different methodologies lead to different interpretations of the tip vortex behavior. Even though the jittering zone is found to be ellipsoidal for all the VIM and schemes, the probability density function
of the vortex center locations varies in the streamwise direction from one single peak with the Graftieaux, vorticity and Q (least-squares scheme) to a binormal distribution with the other implementations.

The multiple peaks, found in some identification parameters, are determined as an artifact produced by certain schemes. The latter can be avoided using either Graftieaux VIM or vorticity and Q-criterion while employing the least-squares scheme.

It is concluded that the vortex center locations are within a small variation range and their comparability is viable inde-
pendently on the VIM or scheme. Nevertheless, first-order schemes, such as backward and forward differences, should be avoided.

The convection velocity presented a higher dependency on the VIM and scheme applied. Therefore, and keeping in mind that the results have shown good comparability regarding the vortex center locations, it is recommended to use the information of several vortex ages instead of the swirling velocity approach to estimate the convection velocity. Conversely, the vortex core
radius only showed a grid step variation between VIMs and schemes. Further studies might include analytical approaches which predict the tangential velocity profiles of a vortex from which is estimated the vortex core to also check their applicability.

Overall, Graftieaux's method is the recommended VIM to track the tip vortex. Indeed, the method does not use differentiation and has shown to be independent of the number of neighboring points used. Moreover, it presents the lowest standard deviation between all the methodologies applied here.

*Data availability.* Velocity field data and results can be provided by contacting the corresponding author.

*Author contributions.* RSV carried out the measurement campaign, worked in the methodology, performed calculations and analysis, and wrote the paper. SC implemented and tested the methodology with assistance and supervision from RSV. SC, SB, MM, CNN, AB and COP contributed with comments and discussions about each section in the manuscript.

*Competing interests.* The authors declare that they have no conflict of interest.

*Acknowledgements.* This research has been supported by the ANID PFCHA/Becas Chile-DAAD/2016 (grant no. 91645539).

# A  Appendix

## A.1  Jittering zones

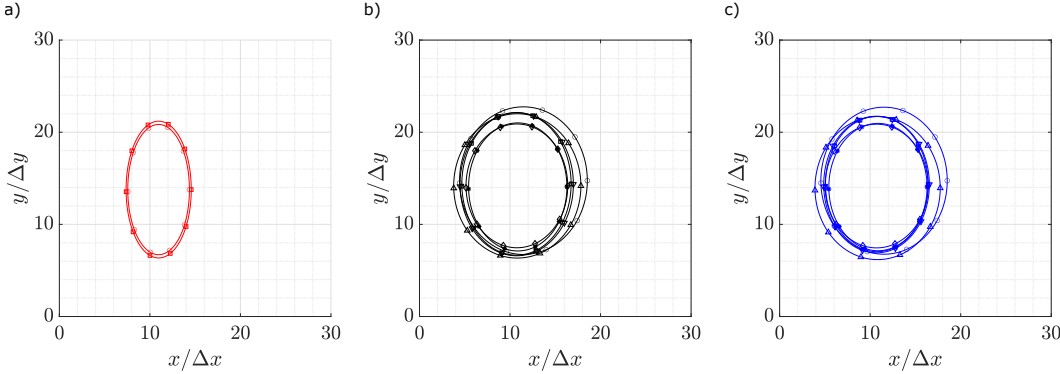

**Figure A1.** Jittering zone of each VIM and scheme. a) Graftieaux. b) Vorticity magnitude. c) Q-criterion.

## A.2  Probability contours

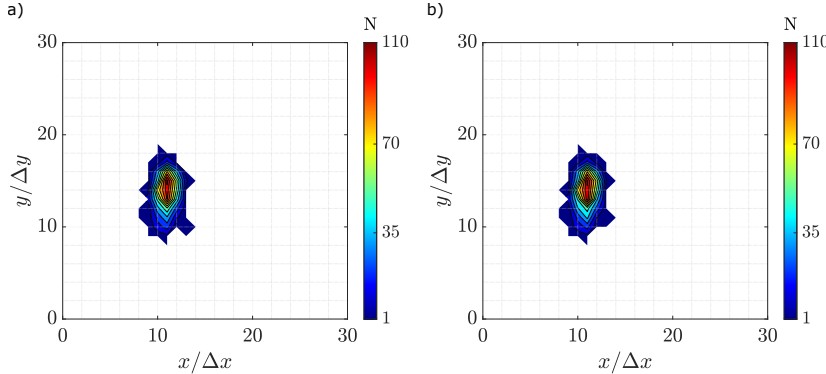

**Figure A2.** Graftieaux probability contours. a) $8-$points. b) $24-$points.

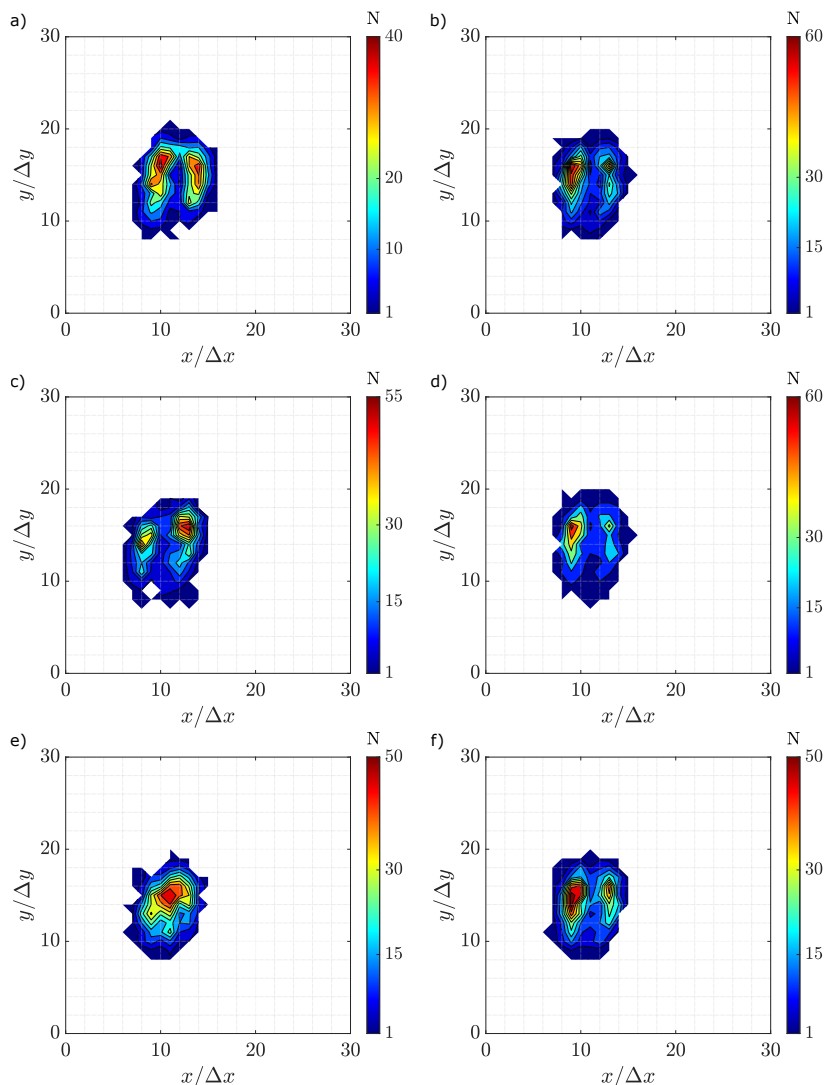

**Figure A3.** Vorticity probability contours. a) Backward difference. b) Central difference. c) Forward difference. d) Richardson extrapolation. e) Least square. f) Circulation method.

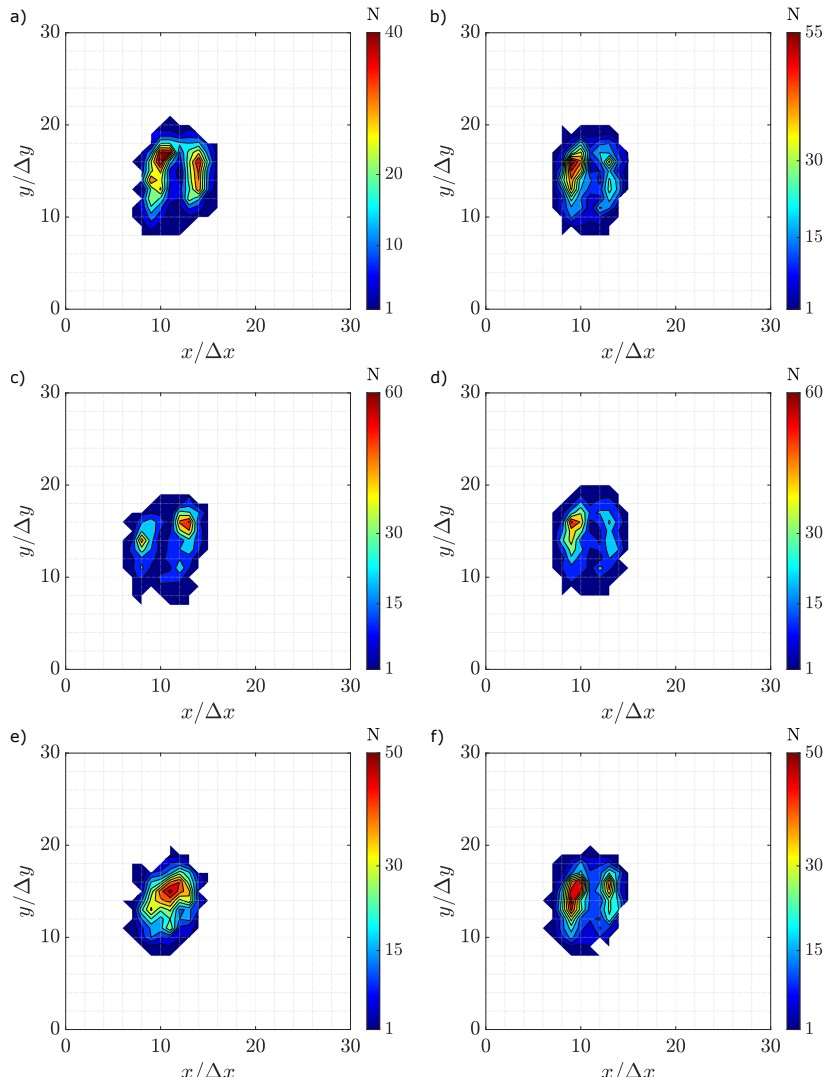

**Figure A4.** Q-criterion probability contours. a) Backward difference. b) Central difference. c) Forward difference. d) Richardson extrapolation. e) Least square. f) Circulation method.

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
