# Peer review of "Vortex identification methods applied to wind turbine tip vortices"

_Wind Energy Science, 2021_

## Referee Comment (RC2)

The authors presented an interesting study in particular in presenting vortex identification methods and their application in PIV data to analyze tip vortex structures of wind turbines. The results are quite informative in terms of identification of vortex core, vortex location and vortex jittering. The manuscript is well designed in structure and clearly presented. To this reviewer, the manuscript is worth being published. However, there are some serious issues need to be further clarified or revised.

General comments:
1- The authors should explain their contribution to the field more clearly. The paper definitely needs more description in terms of its novelty and how it distinguishes itself from previous literatures. In particular, the methods and their applications have been already addressed by other researchers; hence, the authors should demonstrate their contribution.
2- In the introduction section, the authors addressed different PIV measurements performed by previous investigators particularly those focused on tip vortex flow. However, to this reviewer, there are more studies, also worked on the behavior of tip vortices, that can be included in the literature review.
3- The authors have presented an extensive description of VIM methods which predict vortex behaviors. It would be very informative to include analytical approaches such Rankine model which predict tangential velocity profiles of vortex and compare your results with those that can be obtained from those models.
4- The authors are well familiar with the fact that there are two governing parameters, i.e. local Reynolds number and tip speed ratio that affect the flow structure of the turbine including tip vortices. The authors need to discuss further about the role of tip speed ratio in their assessments of vortex location, vortex core radii and vortex jittering.
5- The authors have employed the results obtained from a PIV measurement to perform their analysis. However, they should present more specifications of the PIV test such the sampling rate of the measurement, the phase phase-lock process and the number image pairs per second for each azimuth angle of the blade.
6- The authors should provide enough information if during the measurements the turbine was subjected to any blockage effect in the wind tunnel or not. Regarding that, they should calculate blockage ratio of the turbine based on the tunnel cross section area and considering the tip speed ratio at which the turbine is performing, they should discuss whether the turbine is experiencing blockage effect. If the blockage effect is high, it would affect the experimental results including the velocity field and wake expansion (which also determines the vortex location) significantly.
7- The authors should demonstrate more clearly that how the convection velocity has been estimated, particularly from the PIV data. Did you consider the sampling rate of the measurements for each azimuth angle of the blade? How did you make sure that you are tracking the same vortex as moving from one image pair to the next one?

Specific Comments:
1- Page 3, line 70: what is $theta_M$?
2- Page 7, line 145: what is the tip speed ratio of the turbine? Is it smaller or bigger than the design tip speed ratio?
3- Page 7, line 150: More clarification about the experiment set-up and process is required, such as the sampling rate, frequency of the laser and camera as well as error analysis.

4- Page 7, line 155: How was the process of phase-lock measurements performed?
5- Page 7, Figure 2: the location of the camera is not clear in the figure.
6- Page 10, line 195: what is v(x,y)? it is mentioned that v(x,y) is induced velocity; however, at line 185 induced velocity is represented by u'(x,y). Which one is the correct one? It is confusing.
7- Page 11, line 230: It is mentioned that "the presence of the multiple maxima and the ring-like distribution of the parameters w and Q can be explained through different hypothesis. On one side, the cause could be the level of noise in the vortex core because the lack of seeding." If this can be one of the reasons, why you do not get the similar behavior in Figure 6?

---

## Author Comment (AC1)

**Vortex identification methods applied to wind turbine tip vortices**
Reply to Referee #1

Dear Referee #1, On behalf of all the authors, I would like to thank you for the review of our paper. All your comments will be addressed in the final version, you can find below the reply (blue) to each remark (black).

While the paper provides very interesting insights on the use of the different VIMs for vortex tracking, the novelty of the article is still unclear. The advantages of the Graftieaux's approach, which is hailed by the authors as the most suitable approach are already known. The authors should expand more on the novelty of their article in the introduction section since it is for now not very apparent.

- Reply: We see your point, thank you for highlighting that the novelty was not clear enough. We will clearly state that Graftieaux's method is applied to wind turbine tip vortices for the first time and that the application of the three VIMs to the wind turbine vortices is applied simultaneously with differentiation approaches, which provides explicit comparability in terms of several vortex properties. This will improve the paper and its novelty in the revised submission.

The effects of experimental uncertainty are lacking in the paper. The SPIV data has some level of uncertainty in the results and how this gets propagated in the methods that are being proposed is not very clear. At the very least, a discussion of these effects should be made. This could also have an impact on the Graftieaux approach used. The literature on the uncertainty on PIV should be thoroughly reviewed in order to be able to support your discussion of the effects.

- Reply: Thank you, a section will be added in the final manuscript addressing the uncertainty.

Figure 10 – It is recommended to remove the line plot and use a scatter plot for such representations.

- Reply: Thank you, this will be implemented.

Line 264-267 – The following sentence should be clarified further "Alternative methods such as the prediction from time series vortex locations might be also successful using the rest of the schemes due to the small discrepancy between the vortex center locations between VIMs and schemes; however, more than one vortex age is needed."

- Reply: Thank you, the sentence refers to calculating the convection velocity between two consecutive locations of vortex centers or several consecutive locations by a fitting of the tracking (refer to: Schepers, J. G., & Snel, H. (2007). Model experiments in controlled conditions. ECN Report: ECN-E-07-042. or Soto-Valle, R., Alber, J., Manolesos, M., Nayeri, C. N., & Paschereit, C. O. (2020, September). Wind Turbine Tip Vortices under the influence of Wind Tunnel Blockage Effects. In Journal of Physics: Conference Series (Vol. 1618, No. 3, p. 032045). IOP Publishing.).

> In this study, only one vortex age is analyzed, and the convection velocity is defined as the velocity in the vortex center. The application of the above-referenced methodology (with any VIM), described as "might be successful", is because there are only a couple of grid points discrepancy between VIMs and therefore, the results would be similar independently of the VIM applied.
>
> The sentence will be rephrased in a clearer manner.

Figure 17 – For completeness the figure needs a colorbar.

- Reply: Thank you, this will be implemented.

Line 333 – It is not very clear whether the authors are rejecting the uneven shedding effects on the observed double peak results.

- Reply: Thank you, an explicit sentence on this will be added.

Line 343 – Do you here mean for vortex kinematics analysis or do you really mean that the methods are simply not suitable for establishing both position and motion of the tip vortices? "In fact, both schemes ignore information either forward or backward from the grid on the implementation of differentiation. Therefore, they are not suitable for vortex analysis."

- Reply: Thank you, the reviewer is correct to highlight the strength of this comment. The final sentence will be rephrased and reformulated accordingly within the findings of the work i.e. the large standard deviations.

Conclusion - For the most part it is felt that the issue of the double peak has remained unresolved in this work. The authors seem to attribute these to purely numerical artefacts. Do the authors feel confident about this conclusion? Could experimental uncertainties also be responsible for this?

- Reply: Thank you, this part will be updated accordingly to the uncertainty inclusion.

Line 25 – "It is shown, by using the vorticity to identify the vortices, a high variation in the position of the tip vortices."

- Reply: thank you, it will be rephrased as follow: "A high variation of the position of the tip vortices is shown by using the vorticity in the identification"

Line 237 – Change "are originated" to "might originate"

- Reply: thank you, this will be implemented.

Line 269 – Sentence structure is poor here: "the Graftieaux 24-points as well only vorticity magnitude cases are presented."

- Reply: thank you, it will be rephrased as follow:" [...] In the interest of clarity, only the Graftieaux 24-points and vorticity VIMs are presented [...]"

Line 365 – "The two peaks found in the jittering…" – please rephrase

- Reply: thank you, it will be rephrased as follow: "The two peaks, found in the jittering, are determined as an artifact produced by certain schemes."

  Additionally, this part will be updated accordingly to the uncertainty inclusion.

---

## Author Comment (AC2)

**Vortex identification methods applied to wind turbine tip vortices**
Reply to Referee #2

Dear Referee #2, On behalf of all the authors, I would like to thank you for the review of our paper. All your comments will be addressed in the final version, you can find below the reply (blue) to each remark (black).

The authors should explain their contribution to the field more clearly. The paper definitely needs more description in terms of its novelty and how it distinguishes itself from previous literatures. In particular, the methods and their applications have been already addressed by other researchers; hence, the authors should demonstrate their contribution.

- Reply: We see your point, thank you for highlighting that the novelty was not clear enough. We will clearly state that the methodologies are assessed for the first time simultaneously with several VIMs/schemes, which provides explicit comparability. Additionally, we investigated how these methodologies can impact the definition of the properties of the vortex, such as jittering, convection velocity and core radius, these findings will provide valuable guidelines for future wind tip vortices analysis. This improvement will enhance the paper and its novelty in the revised submission.

In the introduction section, the authors addressed different PIV measurements performed by previous investigators particularly those focused on tip vortex flow. However, to this reviewer, there are more studies, also worked on the behavior of tip vortices, that can be included in the literature review.

- Reply: Thank you. The authors focused the literature review on PIV and tip vortices. It was decided to only include studies where sufficient information about the VIM and the numerical scheme were available. In some cases, where published reports did not include the required information, we contacted the authors and, if the Information was provided, the study was considered. Nevertheless, we agree that the literature review is a significant part of any study, and we would like to have it as complete as possible. We will improve the literature review in the revised manuscript.

-

The authors have presented an extensive description of VIM methods which predict vortex behaviors. It would be very informative to include analytical approaches such Rankine model which predict tangential velocity profiles of vortex and compare your results with those that can be obtained from those models.

- Reply: Thank you. Indeed, it would be very informative, but the referred Rankine model is not the only one [please refer to Rodriguez, S., Espinoza, F., Steinberg, S., & El-Genk, M. (2012). Towards a unified swirl vortex model. In 42nd AIAA Fluid Dynamics Conference and Exhibit (p. 3354).] and including such models would extend the paper in an analytical benchmark direction, which is not the purpose of our research.

The authors are well familiar with the fact that there are two governing parameters, i.e. local Reynolds number and tip speed ratio that affect the flow structure of the turbine including tip vortices. The authors need to discuss further about the role of tip speed ratio in their assessments of vortex location, vortex core radii and vortex jittering.

- Reply: Thank you for your comment. In this work, the tip speed ratio and inflow are fixed. Only one vortex age is analyzed. The Reynolds number based on the circulation is Re=108000 classified as fully turbulent [see Ramasamy, M., & Leishman, J. G. (2006). A generalized model for transitional blade tip vortices. Journal of the American Helicopter Society, 51(1), 92-103]. This will be included in the final version of the manuscript.

*The authors have employed the results obtained from a PIV measurement to perform their analysis. However, they should present more specifications of the PIV test such the sampling rate of the measurement, the phase phase-lock process and the number image pairs per second for each azimuth angle of the blade.
*Page 7, line 145: what is the tip speed ratio of the turbine? Is it smaller or bigger than the design tip speed ratio?
*Page 7, line 150: More clarification about the experiment set-up and process is required, such as the sampling rate, frequency of the laser and camera as well as error analysis.
*Page 7, line 155: How was the process of phase-lock measurements performed?
*Page 7, Figure 2: the location of the camera is not clear in the figure.
* The authors should demonstrate more clearly that how the convection velocity has been estimated, particularly from the PIV data. Did you consider the sampling rate of the measurements for each azimuth angle of the blade? How did you make sure that you are tracking the same vortex as moving from one image pair to the next one?

- Reply: Thank you for pointing out this. In the interest of brevity, we did not include all details in this manuscript, as it is already available in [Soto-Valle, R., Alber, J., Manolesos, M., Nayeri, C. N., & Paschereit, C. O. (2020, September). Wind Turbine Tip Vortices under the influence of Wind Tunnel Blockage Effects. In Journal of Physics: Conference Series (Vol. 1618, No. 3, p. 032045). IOP Publishing.].
  However, we see your point and for completeness, we will include the required details in the revised version.

The authors should provide enough information if during the measurements the turbine was subjected to any blockage effect in the wind tunnel or not. Regarding that, they should calculate blockage ratio of the turbine based on the tunnel cross section area and considering the tip speed ratio at which the turbine is performing, they should discuss whether the turbine is experiencing blockage effect. If the blockage effect is high, it would affect the experimental results including the velocity field and wake expansion (which also determines the vortex location) significantly.

- Reply: Thank you, the turbine experience 40% blockage. In fact, the effects of the blockage were analyzed in previous research work [Soto-Valle, R., Alber, J., Manolesos, M., Nayeri, C. N., & Paschereit, C. O. (2020, September). Wind Turbine Tip Vortices under the influence of Wind Tunnel Blockage Effects. In Journal of

Physics: Conference Series (Vol. 1618, No. 3, p. 032045). IOP Publishing]. Part of the conclusions evidence that the tip vortex travels farther downstream and more inboard compared to simulations and similar experiments with considerably less blockage. Nevertheless, the goal of this research is to study how much the position and characteristics of the vortex change due to the application of different VIMs/ schemes and as such the conclusions from the present work will not be altered by blockage effects.

We will explicitly state this, in the revised manuscript, and propose that the applied VIMs and schemes can be evaluated in a test rig with lesser blockage as further work.

Page 3, line 70: what is ΘM?

- Reply: Thank you for highlighting the missing description. This angle is referred to in Fig. 1. The description will be added to the final manuscript.

Page 10, line 195: what is v(x,y)? it is mentioned that v(x,y) is induced velocity; however, at line 185 induced velocity is represented by u'(x,y). Which one is the correct one? It is confusing.

- Reply: Thank you. You are right, the symbol misses the " ' ". The correct notation is: velocity U=(u,v) and induced velocity U'= (u', v')

Page 11, line 230: It is mentioned that "the presence of the multiple maxima and the ring-like distribution of the parameters ω and Q can be explained through different hypothesis. On one side, the cause could be the level of noise in the vortex core because the lack of seeding." If this can be one of the reasons, why you do not get the similar behavior in Figure 6?

- Reply: Yes, a single peak is visible in Fig. 6. We will add this point in the same paragraph. Part of this led us to the artifact conclusion of some of the schemes applied with vorticity and Q-criterion.

  Multiple peaks because of small structures combined within the core and Grafiteaux's method will be revisited in the final manuscript.

---

## Author Response (AR1)

Dear Editor,

we would like to sincerely thank the Reviewers for their time and constructive feedback regarding the paper "Vortex identification methods applied to wind turbine tip vortices".

We have tried our best to modify the manuscript according to the suggestions. Along with the present document, we are attaching a revised version of the manuscript, where all changes are tracked (wes-2021-104-tracked_changes.pdf) and a clean version of the same manuscript (wes-2021-104-Clean.pdf) with all the changes incorporated.

In the following, Reviewers' comments are reported in black-colored text, while authors' answers are in blue-colored text. Additionally, new sections added in the manuscript that refer to each comment are provided between quotation marks, for convenience.

We believe that Reviewers' comments have helped us making significant improvements in the paper possible. We thus hope that the study can now meet the high standards of WES journal.

Best regards,

*Rodrigo Soto-Valle, Stefano Cioni, Sirko Bartholomay, Marinos Manolesos, Christian Navid Nayeri, Alessandro Bianchini, and Christian Oliver Paschereit*

**Reviewer #1**

**RC1.1** While the paper provides very interesting insights on the use of the different VIMs for vortex tracking, the novelty of the article is still unclear. The advantages of the Graftieaux's approach, which is hailed by the authors as the most suitable approach are already known. The authors should expand more on the novelty of their article in the introduction section since it is for now not very apparent.

**The Reviewer's comment prompted us at trying to better highlight the impact and novelty. A dedicated part has been added to the scope and is reported below for convenience.**

*"Several vortex identification methods (VIMs) have been employed so far. However, consensus on the most suitable methodology for the study of vortices in the wake of a wind turbine has not been found yet, as shown in Table 1. Furthermore, upon examination of the literature, it is apparent that many studies do not provide the complete implementation methodology, such as the differentiation scheme, thus hindering an extensive comparison between methods.*

*This paper aims at comparing different VIMs to evaluate their suitability to study specifically the tip vortices of a wind turbine. The methods are applied to velocity field planes that were obtained through PIV in the near wake of a wind turbine model located in a wind tunnel facility. Compared to previous investigations, the present study offers in a depth comparison, commonly used VIMs on the same wind turbine tip vortex measurement data set. The main goal is to identify similarities and differences of the methodologies, i.e., providing a direct insight into their application. Furthermore, a rigorous comparison of VIM application is provided, with the simultaneous study of six tip vortex parameters, namely: (1) streamwise location; (2) lateral location; (3) streamwise velocity; (4) lateral velocity; (5) core radius; and (6) jittering.*

*Thanks to the large number of analyzed samples, a statistical analysis is also included in order to give more insights into the challenges of each methodology. Three different VIMs are compared:*

*vorticity, Q-criterion and Graftieaux. The first two VIMs require differentiation, thus, the application of six different schemes is examined. Moreover, Graftieaux's methodology is also tested in different scenarios. In this way, a total of 14 cases are presented, where each of the six parameters is investigated. This represents an important source of information to support future wind turbine tip vortices analyses in both experiments and simulations as the implementation is scalable and only requires velocity fields input."*

**RC1.2** The effects of experimental uncertainty are lacking in the paper. The SPIV data has some level of uncertainty in the results and how this gets propagated in the methods that are being proposed is not very clear. At the very least, a discussion of these effects should be made. This could also have an impact on the Graftieaux approach used. The literature on the uncertainty on PIV should be thoroughly reviewed in order to be able to support your discussion of the effects.

**Thank you for the right comment. The discussion about measurement uncertainty has been improved throughout the paper based on three main actions: 1) Table 1 has been updated highlighting the number of samples of each research, i.e., giving an overview of their repeatability; 2) The literature review has been updated including important references about uncertainty. 3) Additional details about experiments have been included, from which the uncertainty levels, based on the error of the velocity field, have been reported. Modifications made on the paper are given below for the completeness of this document.**

*"It is worth remarking that, once comparing the methods the inherent error introduced by the PIV technique must be accounted for. Table 1 includes the number of samples (or pair-samples in Stereo-PIV) used to analyze each contribution. The latter is a well-known parameter, directly related to the uncertainty level. This has been extensively used in literature to give a quantification of the uncertainty in PIV experiments (Grant and Owens, 1990; Micallef, 2012; Del Campo et al., 2014; Micallef et al., 2016), Eq. 1 shows an example of the error in a measured velocity u by Sherry et al. (2013b).*

$$\epsilon_u = \frac{z\, I_u}{\sqrt{N}}, \tag{1}$$

*where z is the confidence coefficient or critical value (normal distribution), I is the turbulence intensity and N is the number of samples. Moreover, it is overall agreed that actions to reduce uncertainty levels could be (1) the maximization of the number of samples to ensure repeatability and convergence of the results (Uzol and Camci, 2001; Ostovan et al., 2019); and (2) the use of subpixel algorithms (Scarano, 2001) giving errors below 0.1 px (Del Campo et al., 2014; Sciacchitano et al., 2013; Beresh, 2008; Fouras and Soria, 1998; Scarano, 2001)"*

**Table 1.** Wind turbine tip vortices studies employing the PIV technique and VIM details.

| contributor | test facility[a] | diameter [m] | N | VIM | scheme |
|---|---|---|---|---|---|
| Whale et al. (2000) | WC | 0.18 | 6 | vorticity magnitude | fifth order polynomial |
| Maalouf et al. (2009) | WT, closed-loop | 0.50 | 95 | circulation | integration |
| Xiao et al. (2011) | WT, open jet | 1.25 | n.s. | vorticity magnitude | n.s. |
| Yang et al. (2011) | WT, closed-loop | 0.25 | 1000 | vorticity magnitude | n.s. |
| Micallef et al. (2014) | WT, open-jet | 2.00 | 100 | vorticity magnitude | central difference |
| Meyer et al. (2013) | WC | 0.38 | 100 | vorticity magnitude | n.s. |
| Sherry et al. (2013a) | WC | 0.23 | 25 | Graftieaux's method | solid-body rotation |
| Sherry et al. (2013b) | WC | 0.23 | 300 | swirling strength criterion | Richarson extrapolation |
| Ning and Yang (2013) | WT, open-jet | 0.25 | 960 | vorticity magnitude | n.s. |
| Jin et al. (2014) | WT, closed-loop | 0.15 | 300 | vorticity | n.s. |
| Ostovan et al. (2019) | WT, open jet | 0.94 | 1000 | zero induced velocity | central difference |
| Soto-Valle et al. (2020) | WT, closed-loop | 3.00 | 1200 | $Q$-criterion | central difference |
| Fontanella et al. (2021) | WT, closed-loop | 2.38 | 100 | vorticity magnitude | central difference |

(a) WT: wind tunnel, WC: water channel. n.s: not specified.

**Table 3.** Operation and image acquisition details.

| operation parameters | | PIV parameters | |
|---|---|---|---|
| cross-section area | $4.2 \times 4.2\ m^2$ | cameras | PCO 2000 |
| BeRT rotor radius | $1.5\ m$ | lens focal length | $100\ mm$ |
| blockage ratio | $40\%$ | resolution | $2048 \times 2048\ px^2$ |
| freestream speed | $6.5\ ms^{-1}$ | field of view | $435 \times 435\ mm^2$ |
| rotational speed | $3\ Hz$ | recordings | 1200 |
| tip speed ratio | 4.35 | laser pulse separation | $150\ \mu s$ |
| turbulence intensity[a] | $3-6\%$ | interrogation window | $24 \times 24\ px^2$ (50% overlapping) |
| phase-locked angle | $\phi = 40°$ | spatial resolution | $3.6\ mm$ |

(a) reported by Bartholomay et al. (2017)

*"Based on the selected operational parameters (see Table 3), the uncertainty of the velocity magnitude is below $\pm 0.4\%$ of the freestream velocity (see Eq. 1), which is equivalent to $0.026 ms^{-1}$ with a $98\%$ confidence. This uncertainty level does not affect the location of the vortex centers of the averaged velocity field or the other studied parameters, as they rely on the vortex center location. For completeness, a statistical analysis of the instantaneous velocity fields is done and is presented in the following section."*

**RC1.3** Figure 10 – It is recommended to remove the line plot and use a scatter plot for such representations.

**The Reviewer's correction is right. Both convection velocity plots were updated using bars instead of a continuous line. The new Figure is reported below for convenience.**

[Figure]

[Figure]

a)      b)

**Figure 10.** Average convection velocities.

**RC1.4** Line 264-267 – The following sentence should be clarified further "Alternative methods such as the prediction from time series vortex locations might be also successful using the rest of the schemes due to the small discrepancy between the vortex center locations between VIMs and schemes; however, more than one vortex age is needed."

**The original paragraph referred to the possibility of calculating the convection velocity between two or more consecutive vortex center locations by means of a fitting curve between either streamwise or lateral locations over time. In particular, the streamwise location of the vortex presents a linear behavior over time. In the revised manuscript this part has been rewritten and two references where this methodology is applied were added. The rephrased paragraph is given below for convenience:**

*"Therefore, the estimation of the convection velocity is recommended with the smoother VIMs and schemes, i.e., the Graftieaux method or vorticity, and Q-criterion while employing LS or CM schemes. Additionally, since there is a low scattering in vortex locations among VIMs and schemes, the convection velocity can be alternatively calculated by comparing several vortex locations over time, fitting streamwise and lateral locations separately (Snel et al., 2007; Soto-Valle et al., 2020). However, more than one vortex age is needed."*

**RC1.5** Figure 17 – For completeness the figure needs a colorbar.

**Figures 9 and 17 were updated with their respective colorbars. Both Figures are shown here for convenience.**

[Figure]

**Figure 9.** Vortex center locations for different differentiation schemes. The vorticity magnitude contour based on the least squares scheme is shown.

[Figure]

**Figure 17.** Instantaneous vorticity magnitude with the central differentiation scheme and quiver lines of the velocity field.

**RC1.6** Line 343 – Do you here mean for vortex kinematics analysis or do you really mean that the methods are simply not suitable for establishing both position and motion of the tip vortices? "In fact, both schemes ignore information either forward or backward from the grid on the implementation of differentiation. Therefore, they are not suitable for vortex analysis."

**The sentence was unclear, and it has been rephrased according to the findings of the work. It is reported below for convenience:**

*"Based on the above and due to the large values of SD after the application of these schemes, they are not recommended for vortex analysis"*

**RC1.7** Line 333 – It is not very clear whether the authors are rejecting the uneven shedding effects on the observed double peak results.

Conclusion - For the most part it is felt that the issue of the double peak has remained unresolved in this work. The authors seem to attribute these to purely numerical artefacts. Do the authors feel confident about this conclusion? Could experimental uncertainties also be responsible for this?

**The two hypotheses of multiple peak results are now in separated paragraphs to make their formulation clearer. The supported hypothesis is now explicitly reported in the manuscript in both the averaged and statistical results. For convenience, the relevant paragraphs are given below. Additionally, the title of Section 5 has been updated to: Results and discussion.**

[revised manuscript text omitted]

**Reviewer #2**

**RC2.1** The authors should explain their contribution to the field more clearly. The paper definitely needs more description in terms of its novelty and how it distinguishes itself from previous literatures. In particular, the methods and their applications have been already addressed by other researchers; hence, the authors should demonstrate their contribution.

**The Reviewer's comment prompted us at trying to better highlight the impact and novelty. A dedicated part has been added to the scope and is reported below for convenience.**

*"Several vortex identification methods (VIMs) have been employed so far. However, consensus on the most suitable methodology for the identification of vortices in the wake of a wind turbine has not been found yet, as shown in Table 1. Furthermore, upon examination of the literature, it is apparent that many studies do not provide the complete implementation methodology, such as the differentiation scheme, thus hindering an extensive comparison between methods.*

*This paper aims at comparing different VIMs to evaluate their suitability to study specifically the tip vortices of a wind turbine. The methods are applied to velocity field planes that were obtained through PIV in the near wake of a wind turbine model located in a wind tunnel facility. The main goal is to provide similarities and differences of the methodologies after being applied to the same dataset, i.e., providing a direct insight into their comparability. The application of the methodology is based on six tip vortex parameters, namely: (1) streamwise location; (2) lateral location; (3) streamwise velocity; (4) lateral velocity; (5) core radius; and (6) jittering.*

*Thanks to the large number of analyzed sample, a statistical analysis is also included in order to give more insights into the challenges of each methodology. Three different VIMs are compared: vorticity, Q-criterion and Graftieaux. The first two VIMs require differentiation, thus, the application of six different schemes is examined. Moreover, Graftieaux's methodology is also tested in different scenarios. In this way, a total of 14 cases are presented, where each of the six parameters is investigated. This represents an important source of information to support future wind turbine tip vortices analyses in both experiments and simulations as the implementation is scalable and only requires velocity fields input."*

**RC2.2** In the introduction section, the authors addressed different PIV measurements performed by previous investigators particularly those focused on tip vortex flow. However, to this reviewer, there are more studies, also worked on the behavior of tip vortices, that can be included in the literature review.

**Thank you for the comment. The literature review has been improved including more studies about tip vortices and uncertainty evaluation. Moreover, Table 1 has been updated, with these studies. One column has been also added to the Table, to display the number of samples used by each of the contributors in their analysis. Both the updated literature and Table 1 are given below for convenience.**

[revised manuscript text omitted]

(a) WT: wind tunnel, WC: water channel. n.s: not specified.

**RC2.3** The authors have presented an extensive description of VIM methods which predict vortex behaviors. It would be very informative to include analytical approaches such Rankine model which predict tangential velocity profiles of vortex and compare your results with those that can be obtained from those models.

**The Reviewer's comment is very interesting and could represent a very engaging continuation of the research. A reference on this has been added to the paper and is given below. Such models would extend, however, the paper in an analytical benchmarking direction, probably lowering the focus on the experimental part.**

*"The convection velocity presented a higher dependency on the VIM and scheme applied. Therefore, and keeping in mind that the results have shown good comparability regarding the vortex center locations, it is recommended to use the information of several vortex ages instead of the swirling velocity approach to estimate the convection velocity. Conversely, the vortex core radius only showed a grid step variation between VIMs and schemes. Further studies might include analytical approaches which predict the tangential velocity profiles of a vortex from which is estimated the vortex core to also check their applicability."*

**RC2.4** Experimental details:

*The authors are well familiar with the fact that there are two governing parameters, i.e. local Reynolds number and tip speed ratio that affect the flow structure of the turbine including tip vortices. The authors need to discuss further about the role of tip speed ratio in their assessments of vortex location, vortex core radii and vortex jittering.

*The authors have employed the results obtained from a PIV measurement to perform their analysis. However, they should present more specifications of the PIV test such the sampling rate of the measurement, the phase phase-lock process and the number image pairs per second for each azimuth angle of the blade.

*Page 7, line 145: what is the tip speed ratio of the turbine? Is it smaller or bigger than the design tip speed ratio?

*Page 7, line 150: More clarification about the experiment set-up and process is required, such as the sampling rate, frequency of the laser and camera as well as error analysis.

*Page 7, line 155: How was the process of phase-lock measurements performed?

*Page 7, Figure 2: the location of the camera is not clear in the figure.

*The authors should provide enough information if during the measurements the turbine was subjected to any blockage effect in the wind tunnel or not. Regarding that, they should calculate blockage ratio of the turbine based on the tunnel cross section area and considering the tip speed ratio at which the turbine is performing, they should discuss whether the turbine is experiencing blockage effect. If the blockage effect is high, it would affect the experimental results including the velocity field and wake expansion (which also determines the vortex location) significantly.

**The authors would like to thank the Reviewer for his/her right suggestion. The following actions have been taken: 1) All additional details on experiments have been added in the paper; 2) A summary Table has been included; and 3) Figure 2 has been updated. All changes are given below for convenience:**

*"The experiments were carried out in the closed-loop wind tunnel at the Technische Universität Berlin. The wind turbine, Berlin Research Turbine (BeRT) (Pechlivanoglou et al., 2015), is a three-bladed, upwind horizontal axis wind turbine model. Blades are twisted, tapered, and based on Clark Y airfoil profile along the full span. Moreover, the blade-tip is sword-shaped and the Reynolds number, based on the circulation of the tip vortices, is $Re_v \approx 10^5$ (Soto-Valle et al. (2020)). The freestream velocity and rotational frequency are fixed giving a tip speed ratio of 4.35, which is the design-rated condition of the turbine. The latter provides a constant operational condition to all the studied vortices. Table 3 reports details of the experimental setup.*

*The wind turbine model produces a 40% blockage ratio in the wind tunnel, while this is quite relevant for performance measurements, it is thought to be acceptable for this study as all the identification methods and schemes are applied to the same dataset and with the focus of highlighting the differences in their outcomes. Therefore, conclusions should not be altered by this effect.*

*The stereo-PIV system consisted of a Quantel Dual-Nd:Yag double laser with energy of 171mJ, a mirror arm, the laser sheet optics and two cameras (CCD-chip). Additionally, an ILA synchronizer receives information of a reference blade azimuthal angle from a light sensor located in the nacelle. In this way, the phase-locked measurement is achieved by coupling the laser and blade position. Table 3 provides details of the PIV system.*

*The measurement plane was horizontal and was centered on the tip location when the blade was in the horizontal position. In this study, only one vortex age is analyzed, $\phi = 40°$, consequently, all the studied parameters belong to the same vortex age, shed from consecutive rotations. A total of 1200 image pairs are recorded in the phase-locked position, this ensures enough information to obtain converged statistics of the results (Uzol and Camci, 2001; Ostovan et al., 2019). The image postprocessing is done with sub-pixel precision by three-point Gaussian fit using the software PIVview3C (PIVTec GmbH). Figure 2 shows a sketch of the facility together with details of the camera and the calibration procedure."*

**Table 3.** Operation and image acquisition details.

| operation parameters | | PIV parameters | |
|---|---|---|---|
| cross-section area | $4.2 \times 4.2\ m^2$ | cameras | PCO 2000 |
| BeRT rotor radius | $1.5\ m$ | lens focal length | $100\ mm$ |
| blockage ratio | 40% | resolution | $2048 \times 2048\ px^2$ |
| freestream speed | $6.5\ ms^{-1}$ | field of view | $435 \times 435\ mm^2$ |
| rotational speed | $3\ Hz$ | recordings | 1200 |
| tip speed ratio | 4.35 | laser pulse separation | $150\ \mu s$ |
| turbulence intensity[a] | $3-6\%$ | interrogation window | $24 \times 24\ px^2$ (50% overlapping) |
| phase-locked angle | $\phi = 40°$ | spatial resolution | $3.6\ mm$ |

(a) reported by Bartholomay et al. (2017)

[Figure]

**Figure 2.** Front view sketch of Berlin Research Turbine ① (BeRT), cameras ② and laser sheet ③, left. Cameras system, middle. BeRT and calibration target in the test section.

**RC2.5** The authors should demonstrate more clearly that how the convection velocity has been estimated, particularly from the PIV data. Did you consider the sampling rate of the measurements for each azimuth angle of the blade? How did you make sure that you are tracking the same vortex as moving from one image pair to the next one?

**To address this comment, subsection 4.2, which contained the methodology for the vortex center and convection velocity, was split. Now, the convection velocity is reported in a separate subsection, which is given below for convenience. Additionally, the experimental details provide information about the recording procedure (see previous comment):**

*"4.2 Convection velocity*

*The tip vortex, after being shed, is both translating and rotating at the same time. Considering this, the convection velocity (downstream, x and outboard directions, y) is estimated as the velocity magnitude corresponding to the vortex center location, Eq. 8. The latter is a common estimation in the literature (van der Wall and Richard, 2006; Yamauchi et al., 1999) and it has the advantage that only one vortex age is needed. However, the estimation is also affected by both the VIM and the scheme chosen on their application."*

**RC2.6** Page 3, line 70: what is ΘM?

**The description has been added and is given below for convenience:**

*"where $P$ is a fixed point to evaluate, $\overrightarrow{U_M}$ is the velocity of the $M$ surrounding points to $P$ in the surface $S$, $\overrightarrow{PM}$ is the radius vector that connects the point $P$ with $M$. $N$ is the total number of points considered in the surrounding of $P$, and $z$ is the unit vector, normal to the surface plane $S$. The angle $\theta_M$ is formed by the vector $\overrightarrow{PM}$ and $\overrightarrow{U_M}$."*

**RC2.7** Page 10, line 195: what is v(x,y)? it is mentioned that v(x,y) is induced velocity; however, at line 185 induced velocity is represented by u'(x,y). Which one is the correct one? It is confusing.

**The notation has been corrected and it is given below for convenience**

*"a) Induced velocity field $v'(x, y)$ b) Swirling velocity with the x-axis shifted to the corresponding vortex center."*

**RC2.8** Page 11, line 230: It is mentioned that "the presence of the multiple maxima and the ring-like distribution of the parameters ω and Q can be explained through different hypothesis. On one side, the cause could be the level of noise in the vortex core because the lack of seeding." If this can be one of the reasons, why you do not get the similar behavior in Figure 6?

**The supported hypothesis is now explicitly reported in the manuscript in both the averaged and statistical results. For convenience, the relevant paragraphs are given below. Additionally, the title of Section 5 has been updated to: Results and discussion.**

*"The presence of multiple maxima and the ring-like distribution of the ω and Q parameters can be explained through different hypotheses. On the one hand, the cause could be the level of noise in the vortex core because of the lack of seeding (Foucaut and Stanislas, 2002; van der Wall and Richard, 2006). The rotational motion of the fluid causes the seeding particles to be pushed at the edges of the vortex. For this reason, the velocity vectors shall be evaluated through interpolation, introducing a further source of uncertainty in the results. In this way, the contours of ω and Q have a single peak concentration for the schemes with the lowest uncertainties (LS and CM) while two peak concentrations appear for the schemes with higher uncertainty (CD, RE, BD and FD).*

*On the other hand, the presence of multiple maxima might also be due to small-scale structures within the vortex, as suggested by Bonnet (1998). It is conceivable that these structures might originate during the shedding of the tip vortex from the blade. Certainly, the pressure difference between the pressure and suction sides of the blade is only one of the many effects that take part in the formation of the tip vortices. Several experiments show that the flow at wingtips involves the interaction of multiple vortices, shear layer instabilities, flow separation and re-attachment (Giuni and Green, 2013a; Devenport et al., 1996; Micallef, 2012). The involved structures are also affected by the blade shape, tip geometry, Reynolds number, and load distribution (Giuni and Green, 2013a) and generally merge into a single structure. In conclusion, the multiple peaks could be caused by the uneven shedding of vorticity in the chordwise direction. In the work of Micallef et al. (2014), a study of the formation of the tip vortices in a horizontal axis wind turbine, a complex vorticity distribution along the blade chord is observed, which seems to cause multiple vorticity peaks inside the core. These multiple peaks can be identified in the vortex core even after the tip vortex has been shed from the blade. In present results, the same effect is obtained when the high uncertainty schemes are applied (CD, RE, BD and FD).*

*Among these hypotheses, the first one seems the most suitable. It is possible that artifacts are produced on some of the schemes applied, where the concentration of seeding is diminished. These artificial peaks are not present in the results using the Graftieaux method because the methodology includes information from a larger amount of grid points.*

*In fact, eight and 24 points are employed to estimate the parameter $\Gamma_1$. In the case of BD, CD and FD only two grid points are considered. RE and LS schemes use four grid points, with the difference that in the first case the inner points are considerably weighted more (see Table 2); the opposite happens for the LS case. CM scheme employs six grid points. Therefore, either weighting more the outer part of the derivative estimation (LS scheme) or considering more grid points (CM scheme) contribute to repairing the artifacts and put in evidence that the issue only occurs in the inner part of the vortex."*